# MiR172-*APETALA2-like* genes integrate vernalization and plant age to control flowering time in wheat

**Juan M. Debernardi**[1,2]*, **Daniel P. Woods**[1,2], **Kun Li**[1,2], **Chengxia Li**[1,2], **Jorge Dubcovsky**[1,2]*

1 Department of Plant Sciences, University of California, Davis, California, United States of America,
2 Howard Hughes Medical Institute, Chevy Chase, Maryland, United States of America

* jmdebernardi@ucdavis.edu (JMD); jdubcovsky@ucdavis.edu (JD)

**Data Availability Statement:** All the raw data supporting every figure and every table is provided as Supplemental Data.

## Abstract

Plants possess regulatory mechanisms that allow them to flower under conditions that maximize reproductive success. Selection of natural variants affecting those mechanisms has been critical in agriculture to modulate the flowering response of crops to specific environments and to increase yield. In the temperate cereals, wheat and barley, the photoperiod and vernalization pathways explain most of the natural variation in flowering time. However, other pathways also participate in fine-tuning the flowering response. In this work, we integrate the conserved microRNA miR172 and its targets *APETALA2*-like (*AP2L*) genes into the temperate grass flowering network involving *VERNALIZATION 1* (*VRN1*), *VRN2* and *FLOWERING LOCUS T 1* (*FT1* = *VRN3*) genes. Using mutants, transgenics and different growing conditions, we show that miR172 promotes flowering in wheat, while its target genes *AP2L1* (*TaTOE1*) and *AP2L5* (*Q*) act as flowering repressors. Moreover, we reveal that the miR172-*AP2L* pathway regulates *FT1* expression in the leaves, and that this regulation is independent of *VRN2* and *VRN1*. In addition, we show that the miR172-*AP2L* module and flowering are both controlled by plant age through miR156 in spring cultivars. However, in winter cultivars, flowering and the regulation of *AP2L1* expression are decoupled from miR156 downregulation with age, and induction of *VRN1* by vernalization is required to repress *AP2L1* in the leaves and promote flowering. Interestingly, the levels of miR172 and both *AP2L* genes modulate the flowering response to different vernalization treatments in winter cultivars. In summary, our results show that conserved and grass specific gene networks interact to modulate the flowering response, and that natural or induced mutations in *AP2L* genes are useful tools for fine-tuning wheat flowering time in a changing environment.

## Author summary

Reproductive success is essential for species survival, and in cultivated crops to maximize yield. Plants can sense and integrate different internal and environmental signals to ensure that flowering occurs under optimal conditions. In the temperate cereals, wheat and barley, specific mechanisms have evolved that guarantee flowering is promoted by the longer

**Funding:** J.D. acknowledges support from the Howard Hughes Medical Institute (HHMI Researcher Funding, https://www.hhmi.org/) and by competitive Grants 2022-67013-36209 and 2022-68013-36439 (WheatCAP) from the United States Department of Agriculture, National Institute of Food and Agriculture (https://nifa.usda.gov/). J. M.D. was supported by a fellowship (LT000590/ 2014-L) of the Human Frontier Science Program (hfsp.org). D.P.W is a Howard Hughes Medical Institute Fellow of the Life Sciences Research Foundation (http://www.lsrf.org/). The funders had no role in study design, data collection and analysis, decision to publish, or preparation of the manuscript.

**Competing interests:** The authors have declared that no competing interests exist.

days of spring only after the plants have been exposed to the cold days of winter, a process called vernalization. In this work, we characterized the interactions between the vernalization requirement and a conserved pathway that integrates plant age into flowering regulation. This pathway involves the sequential action of two microRNAs, miR156 and miR172. In spring wheat cultivars, miR156 expression decreases with plant age, while miR172 expression increases. This results in the downregulation of its targets, the *APETALA2*-like (*AP2L*) flowering repressors, and the induction of flowering. In winter wheat cultivars, however, the induction of miR172 and the downregulation of *AP2L1* is decoupled from miR156, and induction of the *VERNALIZATION1* gene by vernalization is required to repress *AP2L1* and promote flowering. Our results show that natural or induced mutations in the *AP2L* genes are useful tools for fine-tuning wheat flowering time in a changing environment.

## Introduction

The precise control of flowering is central to plant reproductive success, and in cultivated cereals to maximize grain yield. Plants have evolved mechanisms that integrate various endogenous and environmental signals such as changes in day-length and temperature that enable them to flower under conditions that optimize seed production. Flowering takes place during a particular time of the year in response to perception of seasonal cues, such as changes in the length of the day or the night by a process called photoperiodism [1]. Temperate grasses, which include the agronomically important crops wheat (*Triticum aestivum* L.) and barley (*Hordeum vulgare* L.), flower in the spring or early summer in response to shorter nights (longer photoperiods) and are referred to as long-day (LD) plants (e.g. [2,3]). Many plants adapted to temperate climates also have a winter annual or biennial life cycle strategy [4]. These plants become established in the fall, overwinter, and flower rapidly in the spring. Essential to this adaptive strategy is that flowering does not occur prior to winter, during which flowering would not lead to successful reproduction. Thus, these plants have evolved regulatory mechanisms to prevent fall flowering and sense the passing of winter to establish competence to flower [5]. The process by which flowering is promoted by a long exposure to cold temperatures is known as vernalization [5].

Comparisons of flowering regulatory networks in different plant species revealed several conserved genes and gene families, but also clade specific genes [6–8]. Different plant clades have evolved flowering network architectures that include novel components, extensive variation in gene expression patterns and/or interactions among conserved genes. Still, a common feature of these networks is that they converge on a small number of floral integrator genes that initiate the early stages of flowering [6].

In the temperate grasses, vernalization and photoperiod pathways converge in the transcriptional activation of *FLOWERING LOCUS T* (*FT*)-like genes in the leaves [1,9] and allelic variation at the *FT1* locus in both barley and wheat is responsible for differences in heading time [9–11]. The central role of *FT*-like genes in initiating flowering appears to be conserved across flowering plants [2,12], and the timing of flowering depends largely on changes in *FT* expression in leaves. *FT* encodes a mobile protein that travels from the leaves to the shoot apical meristem (SAM) [13,14] where it interacts with the bZIP transcription factor FD to activate floral genes and transform the vegetative meristem into a floral meristem [15–18].

The major determinant of the photoperiodic regulation of *FT1* in wheat and barley is *PHOTOPERIOD1* (*PPD1* or *PRR37*) [19]. *PPD1* encodes a protein with a pseudo-receiver domain

and a CONSTANS, CONSTANS-like, TIMING OF CAB EXPRESSION 1 (CCT) domain that promotes *FT1* expression under LD conditions [19]. *PPD1* is the result of a grass specific duplication event (*PRR37/PPD1* and *PRR73*) that is independent of the *PRR3-PRR7* duplication in *Arabidopsis thaliana* (Arabidopsis) [20]. The two Arabidopsis genes are part of the circadian clock, but in the temperate grasses *PPD1* has a more specialized role in the photoperiod pathway [19,21]. *PPD1* expression in the leaves is regulated by the circadian clock and phytochrome-mediated light signaling pathways, ensuring that PPD1 can only promote *FT1* expression under LD conditions [3,19,21–24].

Mutations in the *PPD-H1* coding region in barley result in non-functional or hypomorphic alleles with reduced ability to induce flowering under LD [19]. By contrast, deletions in the promoter regions of *PPD-A1* (*Ppd-A1a* allele) and *PPD-D1* (*Ppd-D1a* allele) homeologs in wheat result in increased *PPD1* expression and accelerated heading under short day (SD) relative to the ancestral *Ppd1b* allele [25,26]. As a result, plants carrying the *Ppd1a* alleles show reduced photoperiod response and are designated as photoperiod insensitive (PI), whereas plants carrying the *Ppd1b* allele are designated as photoperiod sensitive (PS) [25,26]. In addition to *PPD1*, temperate grasses have a parallel photoperiod pathway that involves *CONSTANS* (*CO*)-like genes, *CO1* and *CO2* [22]. However, while in other species *CO*-like genes play a dominant role in photoperiod perception [6,12], in wheat and barley *CO*-like genes have a limited effect that is more relevant in the absence of *PPD1* or when *PPD1* has reduced function [22,27,28].

In wheat, *FT1* expression is also regulated by the vernalization pathway. A current molecular model of vernalization in temperate grasses consists of three large-effect loci acting in a regulatory loop including *VRN1*, *VRN2* and *FT1* (also known as *VRN3* in wheat, [9]). *VRN2*, which is a LD flowering repressor, antagonizes the role of *PPD1* as a LD flowering promoter, preventing the induction of *FT1* in the leaves during the fall [29]. *VRN2* encodes a protein containing a zinc finger motif and a CCT domain and is orthologous to rice GHD7 [30,31] but has no known orthologs in eudicots [29,31]. Wheat and barley plants harboring loss-of-function *VRN2* alleles display a spring growth habit [29,32–35]. *VRN1* encodes a MADS box transcription factor of the SQUAMOSA clade [36,37], and its expression in leaves and apices is induced by vernalization [36]. Winter wheat varieties with a functional but recessive *vrn1* allele require several weeks of vernalization to induce *VRN1* and acquire competence to flower [36,38,39]. In contrast, spring wheat varieties carrying dominant *Vrn1* alleles, which are expressed in the absence of cold temperature, bypass the vernalization requirement [38,40,41].

The expression of *VRN1* in the leaves promotes the repression of *VRN2* [41]. Thus, at the onset of spring as day length is extended, *VRN2* expression levels are low due to the induction of *VRN1* by the previous winter's cold temperatures allowing PPD1 to induce the expression of *FT1* in the leaves. In addition, a positive feedback loop between *VRN1* and *FT1* results in elevated levels of *VRN1* in the presence of elevated *FT1* levels and vice versa [42–44]. Therefore, in winter plants the repression of *FT1* by VRN2 in the fall guarantees that promotion of flowering by long days is blocked until plants have been exposed to winter temperatures, and the positive feedback loop in the spring secures the commitment to flowering (reviewed in [43]).

In addition to the major photoperiodic and vernalization genes, additional genes and pathways integrate multiple signals that impact flowering, such as the circadian clock, the nutritional and developmental status of the plant and a variety of biotic and abiotic stressors [7,45–51]. For example, members of the APETALA2-like (AP2L) family (euAP2 lineage) of transcription factors have been shown to act as flowering repressors [52,53]. These transcription factors, defined by the presence of two AP2-like domains in tandem, are highly conserved in plants. They play important roles in the regulation of the flowering transition by integrating information related to the age of the plant and environmental signals [54].

In Arabidopsis, AP2L transcription factors repress *FT* transcription in the leaves as well as other flowering activator genes in the SAM of juvenile plants [55–57]. In addition, AP2L proteins interact with CO-like proteins and inhibit CO activity [57]. As a result, Arabidopsis plants with mutations in multiple *AP2L* members are rapid flowering under both LD and SD conditions [56,58,59]. Interestingly, in the perennial species *Arabis alpina* (*A. alpina*), a close relative of Arabidopsis, the *AP2L* gene *PERPETUAL FLOWERING2* (*PEP2*) is a flowering repressor that controls the vernalization response and contributes to the perennial life cycle [60,61]. The function of *AP2L* genes as flowering repressors is also conserved in monocot species. In maize (*Zea mays*), overexpression of the *AP2L* genes *Glossy15* and *ZmTARGET OF EAT1* (*ZmTOE1* or *Related to APETALA2.7*, *ZmRAP2*.7) delays flowering [62,63]. In rice (*Oryza sativa*), the *AP2L* genes *INDETERMINATE SPIKELET 1* (*OsIDS1*), *SUPERNUMERARY BRACT* (*SNB*) and *OsTOE1* also act as flowering repressors [64,65]. In wheat, loss-of-function mutations in the domestication gene *Q* (also named *AP2L5* = wheat ortholog of *IDS1*) accelerate flowering [66,67].

The expression of *AP2L* genes is regulated at the post-transcriptional level by miR172, which is an ancient and conserved miRNA in plants [54]. Across flowering plant diversification, the expression of miR172 increases during and after the transition to flowering [58,59,62,64,68–71]. In reproductive tissues, miR172 controls *AP2L* expression to regulate inflorescence and flower development [72,73]. In wheat inflorescences, this regulation is important to control spikelet density, floret number, and the free-threshing character of the spike in domesticated wheat [66,74].

The repression of *AP2L* genes by miR172 is also important to control the timing of the flowering transition. Ectopic expression of *MIR172* genes phenocopies the rapid flowering of mutants in multiple *AP2L* genes in Arabidopsis, rice and wheat [56,58,64–66]. On the other hand, plants with reduced activity of miR172, including Arabidopsis CRISPR mutants in multiple miR172 loci [70,71], plants expressing artificial target mimics against miR172 (henceforth MIM172) or miR172-resistant versions of *AP2L* genes [56,64–66,75], all display a delayed flowering phenotype.

The expression of miR172 in vegetative tissues is controlled by the environment and the developmental status of the plant [58,59,62,64,68,76–78]. As plants age, the expression levels of miR172 increase. This is part of a conserved network that involves another conserved miRNA named miR156, which controls the juvenile to adult vegetative phase transition and flowering competence in several species [76,78]. In this so-called plant age pathway, miR156 is expressed at high levels in juvenile stages and represses the expression of a group of *SQUAMOSA-PROMOTER-BINDING*-like (*SPL*) genes [79]. As plants grow and in response to the increase in sugar levels [80,81], miR156 expression goes down resulting in an increased expression of *SPLs*, which in turn activate the expression of miR172 in the leaves. The increased levels of miR172 result in repression of *AP2L* genes expression, which promotes adult character traits in leaves and also flowering competence [78,82,83].

Our previous studies focused on the role of miR172 and *AP2L* genes in wheat spike development [66,84]. In this work, we focus on the role of the miR172-*AP2L* module in the regulation of the flowering transition through its interaction with the temperate grasses specific flowering pathway involving the *VRN1*-*VRN2*-*FT1* genetic feedback loop. Using a combination of mutants, transgenics, and different growth conditions, we show that miR172 promotes the flowering transition in wheat, while its targets *AP2L* genes work as repressors of this transition. In addition, we show that miR172 and *AP2L* genes regulate the expression of *VRN1*-*VRN2*-*FT1* genes in leaves and modulate the flowering response in spring and winter wheats under different environmental conditions. Finally, we describe mutations in these genes that could be useful tools to fine-tune wheat flowering time in a changing environment.

## Results

### miR172 promotes the flowering transition in spring wheat

To study the role of miR172 in the control of the flowering transition in wheat, we first analyzed transgenic lines with altered levels of miR172 that were previously generated in the tetraploid wheat variety Kronos (*Triticum turgidum* subsp. *durum*) [66] (Fig 1A). Kronos has a spring growth habit determined by the *Vrn-A1c* allele (deletion in intron 1), and a reduced photoperiod response conferred by the *Ppd-A1a* allele [41]. We selected and compared independent $T_2$ transgenic lines overexpressing either miR172 (*UBI_{pro}:miR172*) or a target mimic against miR172 (MIM172). Quantification of miR172 levels by qRT-PCR in a fully expanded fifth leaf confirmed a 4-fold increase in miR172 expression levels in *UBI_{pro}:miR172* plants compared to wild type, and a 20-fold reduction in MIM172 transgenic plants (Fig 1B and Data A in S1 Data).

Under LD conditions (16 h light / 8 h dark), plants overexpressing miR172 headed 4 days earlier (Fig 1C) and produced 0.6 fewer leaves than the wild type (Fig 1D), whereas MIM172 plants headed 8 days later and produced 1.4 more leaves than the wild type (Fig 1C and 1D). Under SD conditions (8 h light / 16h dark) Kronos plants headed in approximately 80 days (S1A and S1B Fig). *UBI_{pro}:miR172* plants flowered earlier than the wild type, and MIM172 plants flowered later than the wild type, similarly to the LD conditions. However, the differences in heading time between *UBI_{pro}:miR172* and MIM172 plants were larger under SD (40 days and 5 leaves, S1A–S1C Fig) than under LD (12 days and two leaves, Fig 1C and 1D). These results show that in spring wheat, miR172 accelerates the transition to flowering both under LD and SD conditions.

### Mutations in *AP2L* genes lead to an acceleration of flowering

We next explored the role of AP2L transcription factors, which are known targets of miR172 in the regulation of flowering time [54]. In wheat, we previously identified four *AP2L* genes with miR172 target sequences (named *AP2L1*, *AP2L2*, *AP2L5*, and *AP2L7*; [66,84]; S1 Table). Loss-of-function *AP2L5* mutants (Fig 1E) showed an early flowering phenotype [66,84] (Fig 1F), whereas the heading time of a null mutant for *AP2L2* was not different to the wild type control [84]. We identified another *AP2L* gene (named *AP2L1*), which is orthologous with *TOE1* (*TARGET OF EARLY ACTIVATION TAGGED (EAT) 1*), a known flowering repressor in Arabidopsis and maize [58,63,84]. Therefore, in this work we further characterized the function of *AP2L1*. We identified TILLING lines with loss-of-function mutations in both the A (K863) and B (CAD0161) homeologs and crossed them to generate an *ap2l1*-null mutant in Kronos (Fig 1E, see Material and Methods). Under LD conditions, the *ap2l1*-null plants headed 5 days earlier than both wild type control and lines harboring mutations in only one *AP2L1* homeolog (Fig 1F). The acceleration of flowering was similar to an *ap2l5*-null mutant grown in the same chamber. We also generated and tested *ap2l5 ap2l1* combined null mutant plants in the same growing conditions. The combined null mutant headed significantly earlier than each of the single gene null mutants and 10 days earlier than the wild type (Fig 1F). These results suggest overlapping and additive roles for these two miR172-targets in the control of flowering time in spring wheat. The *ap2l1*-null mutant did not show any of the spikelet or floret phenotypes previously described for the *ap2l5*-null mutant [66,84] (S2 Fig).

### miR172 promotes flowering by repressing *AP2L* genes

To study the effect of miR172 on *AP2L1* and *AP2L5*, we first checked the expression of these genes in the fifth leaves of Kronos, MIM172 and *UBI_{pro}:miR172* plants, using the same leaf

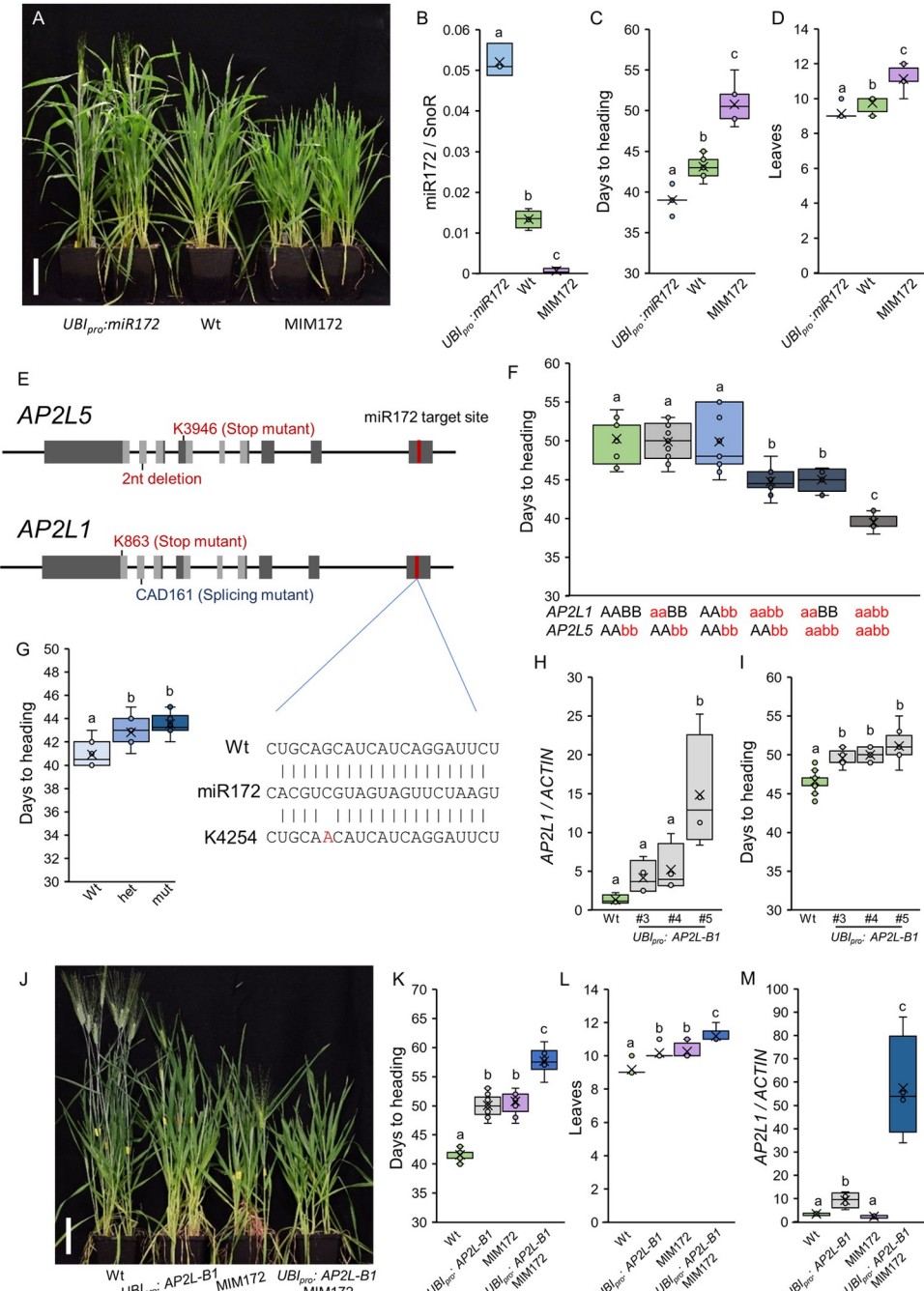

**Fig 1. miR172 promotes flowering in wheat by repressing *AP2L* genes.** (**A**) Six-week-old wild type Kronos plants (Wt) compared to *UBI_{pro}:miR172* and MIM172 transgenic plants in the same genetic background grown under LD. Scale bar = 10 cm. (**B**) Box plots showing miR172 expression levels in the 5th leaf of *UBI_{pro}:miR172*, wild type Kronos (Wt) and MIM172 plants grown under LD. Mature miR172 levels were determined by qRT-PCR using the small nucleolar RNA 101 (SnoR) as internal reference. Data correspond to four independent biological replicates. (**C-D**) Box plots showing days to heading (**C**; n ≥ 7) and number of leaves produced by the main tiller (**D**; n ≥ 7) in the same genotypes and conditions as in B. (**E**) Schematic diagrams showing the gene structures of *AP2L5* and *AP2L1*. Exons are indicated as dark grey boxes, *AP2* domains are indicated in light grey, and the miR172-target sites in red. The positions of the TILLING mutations K3946 (*AP2L5*) and K863 (*AP2L1*) in the A homeologs are indicate above the gene structures, and the natural variant with 2 nucleotides deletion (*AP2L5*) and CAD161 splicing mutant (*AP2L1*) in the B homeologs below. (**F**) Box plots showing days to heading under LD conditions for wild type (Wt), plants harboring mutations in only one *AP2L1* homeolog (aaBB and AAbb), in both *AP2L1* homeologs (*ap2l1*-null = aabb), *ap2l5*-null mutants, and *apl2l1 ap2l5* combined null mutants (n ≥ 8). (**G**) Box plots showing days to heading under LD

conditions for an $F_2$ population segregating for the K4254 mutation in the miR172-target site of *AP2L-A1* gene (n $\geq$ 10).; Wt = homozygous wild type plants, mut = homozygous K4254 mutants, het = heterozygous plants. The interaction between the wild type miR172-target site and miR172 is shown in the right. The TILLING line K4254 has a G>A mutation in the miR172 target site that reduces the free energy of interaction with miR172. (**H-I**) Box plots showing the expression levels of *AP2L1* in the 5th leaves (**H**; n = 4) and days to heading (**I**; n $\geq$ 5) for wild type Kronos (Wt) and three independent *UBI_pro*:*AP2L-B1* transgenic lines grown under LD. (**J-M**) Wild type Kronos (Wt), *UBI_pro*: *AP2L1* (transgenic line #5), MIM172 and MIM172 *UBI_pro*:*AP2L-B1*#5 plants grown under LD. (**J**) Eight-week-old plants. Scale bar = 10 cm. (**K-M**) Box plots showing days to heading (**K**; n $\geq$ 6), number of leaves produced by the main tiller (**L**; n $\geq$ 5) and expression levels of *AP2L1* in the 5th leaves (**M**). Different letters above the box plots indicate significant differences based on Tukey tests ($P < 0.05$), except for panels (**D**), (**F**) and (**L**) where non-parametric Kruskal-Wallis tests were used (Data A in S1 Data).

samples that showed highly significant differences in miR172 expression in Fig 1B. Interestingly, we did not observe differences in the expression of *AP2L1* between MIM172, wild type and *UBI_pro*:*miR172* samples (S3A Fig), and the expression of *AP2L5* was higher in MIM172 compared to wild type but no significant differences were observed between wild type and *UBI_pro*:*miR172* (S3B Fig and Data I in S1 Data). Similar observations were described in other species, where steady-state transcript levels of *AP2L* genes remained constant upon changes in miR172 levels [58,72,85]. Those observations suggested that miR172 could control *AP2L* expression at the translational level [58,72] or that the AP2L proteins could regulate their own transcript levels through feedback loops [55,85].

We further characterized the role of miR172 and *AP2L1* in flowering by studying a Kronos TILLING line (K4254) with a point mutation in the miR172 target site of the *AP2L-A1* homeolog (Fig 1G). This mutation lies in the same position within the miR172 target site as the mutation that generates the dominant *Tasselseed6* allele of *IDS1* in maize [68] and is expected to reduce the interaction with miR172. We backcrossed K4254 two times with wild type Kronos and analyzed heading time under LD in a $BC_1F_2$ segregating population. Plants carrying this dominant mutation headed 3.25 days later than the sister wild type plants (Fig 1G and Data A in S1 Data). This result is consistent with *AP2L1* being a flowering repressor targeted by miR172.

We confirmed the flowering repression activity of *AP2L1* using transgenic plants constitutively expressing the wild type coding region of the *AP2L-B1* gene under the maize *UBIQUI-TIN* promoter (Fig 1H). Three independent *UBI_pro*:*AP2L-B1* lines showed significant delays in heading time compared to the non-transgenic sister lines (2.9–4.5 days, Fig 1I). We then selected the line *UBI_pro*:*AP2L-B1*#5 with the highest *AP2L1* transcript levels (Fig 1H) and crossed it with MIM172, which has reduced miR172. In the resulting $F_2$ population grown under LD conditions (Fig 1J), plants harboring *UBI_pro*:*AP2L-B1*#5 or MIM172 alone headed later and produced more leaves than the wild type (Fig 1K and 1L). Plants carrying both transgenes (*UBI_pro*:*AP2L-B1* and MIM172) showed a larger delay in heading (16.1 days) and produced two more leaves on the main tiller compared to the wild type control, and were significantly later than the individual transgenic lines (Fig 1K and 1L).

Analysis of *AP2L1* expression levels in these lines showed that plants carrying only the MIM172 transgene did not show differences in *AP2L1* expression compared to wild type (as observed in S3A Fig), but plants harboring both transgenes have the highest *AP2L1* transcript levels (Fig 1M).We hypothesize that the expression of *AP2L-B1* under the constitutive *UBIQUITIN* promoter likely disrupted the putative feedback loop of the AP2L1 protein on its own expression, and revealed the activity of miR172 on the *AP2L1* transcript levels. Taken together, these results support the role of miR172 as a repressor of *AP2L1* and a promoter of flowering.

## miR172 and *AP2L* genes regulate flowering by modulating the expression of *VRN1*, *VRN2* and *FT1* in the leaves

We then studied how miR172 and *AP2L* interact with the wheat flowering network involving *PPD1* and the *VRN1-VRN2-FT1* regulatory loop. To this end, we performed a time-course expression analysis using leaves collected from Kronos wild type, $UBI_{pro}$:*miR172* and MIM172 transgenic plants at different ages grown under LD. In the three different genotypes, the expression of flowering promoters *VRN1* and *FT1* (Fig 2A and 2B) increased with plant age, from juvenile leaf 1 (L1) to adult leaf 7 (L7), while the expression of the flowering repressor *VRN2* decreased as plants approached flowering (Fig 2C). In the fifth leaf, we observed significantly higher expression of *VRN1* (Fig 2A) and *FT1* (Fig 2B) in the early flowering $UBI_{pro}$: *miR172* plants than in the wild type. By contrast, in the same leaf, we observed significantly lower expression of *VRN1* (Fig 2A) and *FT1* (Fig 2B) in the late flowering MIM172 plants than in the wild type. A similar expression pattern was observed for the two *VRN1*-related *SQUA-MOSA MADS-box* genes *FUL2* and *FUL3*, whose expression was higher in the fifth leaf of $UBI_{pro}$:*miR172* and lower in MIM172. For these two genes, the differences between wild type and MIM172 were not significant in L5, but were significant in L7 (S4 Fig and Data J in S1 Data). In contrast, the expression levels of *PPD1* were similar among leaves of different ages, and no significant differences with the wild type were observed for $UBI_{pro}$:*miR172* or MIM172 (Fig 2D).

We also checked the expression of these genes in the third and sixth leaves of plants grown under SD conditions (S1D–S1G Fig). The results were similar to LD, with $UBI_{pro}$:*miR172* plants expressing significantly higher levels of *FT1* and *VRN1* in both the third and sixth leaves, lower levels of *VRN2* (only L6), and no significant changes in *PPD1* expression compared to wild type Kronos (Data H in S1 Data). The MIM172 transgenic plants showed similar levels of *VRN2* compared to wild type in both leaves, and lower levels of *VRN1* and *FT1* in the sixth leaf, although the differences were not significant (S1D–S1G Fig). Taken together, these results indicate that constitutive expression of miR172 activity in leaves under both LD and SD conditions is associated with increased levels of the flowering promoters *VRN1* and *FT1*, and decreased levels of the flowering repressor *VRN2*.

We also analyzed the expression of *VRN1*, *VRN2*, and *FT1* in the fifth leaf of plants from an $F_2$ population segregating for $UBI_{pro}$:*AP2L-B1* and MIM172 grown under LD (S5 Fig). In agreement with their late flowering phenotype, the $UBI_{pro}$:*AP2L-B1* MIM172 plants showed the largest reduction in *VRN1* and *FT1* expression compared with the wild type (S5A and S5B Fig and Data K in S1 Data), while the expression of the flowering repressor *VRN2* was significantly increased (S5C Fig). The transgenic plants with either $UBI_{pro}$:*AP2L-B1* or MIM172 showed intermediate values for all three genes. These results indicate that *AP2L* genes repress wheat flowering likely by controlling the expression of these three central wheat flowering genes.

## miR172 regulates *FT1* expression in the absence of *VRN1* and *VRN2*

From the previous experiments, it was not possible to determine which of the three genes is affected by the miR172-*AP2L* module because *VRN1-VRN2-FT1* are interconnected through feedback regulatory loops. To answer this question, we first crossed the $UBI_{pro}$:*miR172* line with *vrn2* loss-of-function Kronos mutant plants [32] and analyzed the flowering phenotype in the resulting $F_2$ population. Under LD conditions, $UBI_{pro}$:*miR172* lines phenocopy the rapid flowering of the *vrn2* mutant, both in days to heading and leaf number (Fig 2E–2G). Moreover, the $UBI_{pro}$:*miR172 vrn2* plants showed an additive early phenotype as compared to the single $UBI_{pro}$:*miR172* or *vrn2* plants (Fig 2E–2G). We collected tissue from the first leaves of these plants to check early responses in the expression of flowering genes. We were not able to

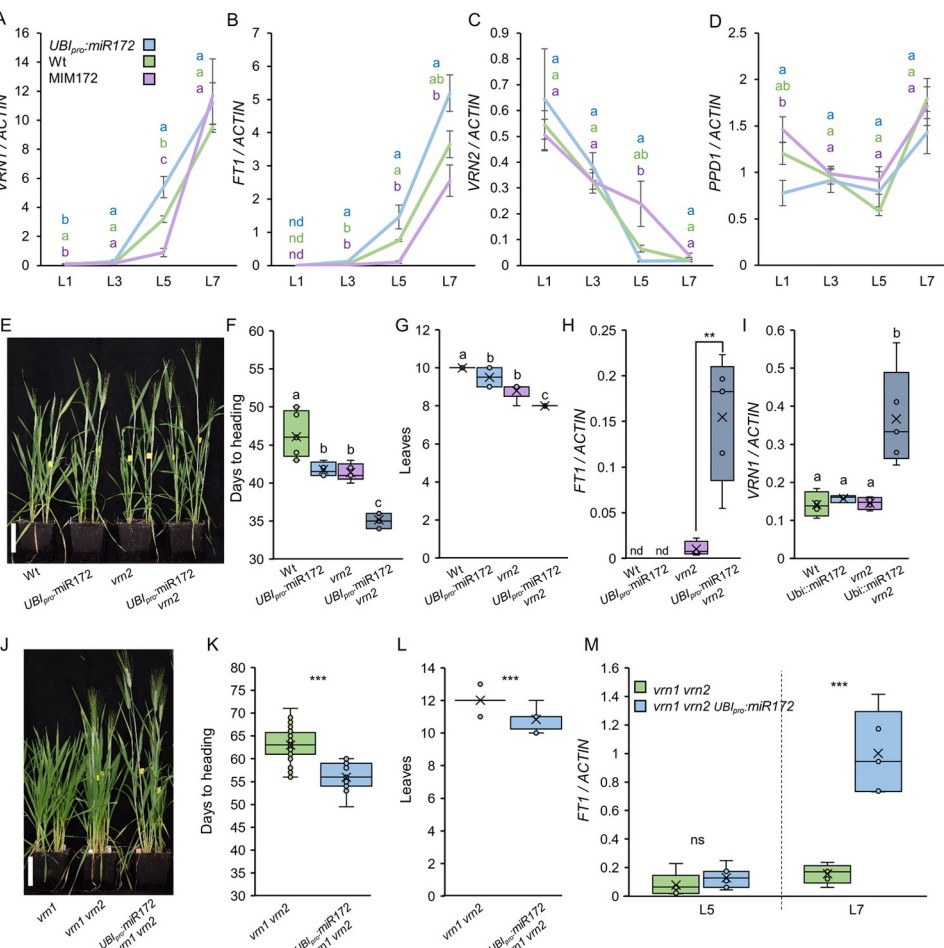

**Fig 2. miR172 controls flowering through the regulation of *FT1* expression in leaves. (A-D)** Expression levels of *VRN1* (**A**), *FT1* (**B**), *VRN2* (**C**), and *PPD1* (**D**) determined by qRT-PCR in the 1st (L1), 3rd (L3), 5th (L5) and 7th (L7) leaves of *UBI_{pro}:miR172*, wild type Kronos (Wt) and MIM172 plants grown under LD. *ACTIN* was used as internal reference. Data correspond to four independent biological replicates. (**E-I**) Population segregating for wild type Kronos (Wt), *UBI_{pro}:miR172*, *vrn2* and *UBI_{pro}:miR172 vrn2* plants grown under LD. (**E**) Six-week-old plants. Scale bar = 10 cm. (**F-G**) Box plots showing days to heading (**F**; n ≥ 4) and total number of leaves produced by the main tiller (**G**; n ≥ 4). (**H-I**) Box plots showing *FT1* (**H**) and *VRN1* (**I**) expression levels determined by qRT-PCR in the first leaves. *ACTIN* was used as internal reference. Data correspond to four independent biological replicates. (**J**) Nine-week-old *vrn1* null, *vrn1 vrn2* combined mutant and *vrn1 vrn2 UBI_{pro}:miR172* plants growing under LD. Scale bar = 10 cm. (**K-M**) *vrn1 vrn2* combined mutant and *vrn1 vrn2 UBI_{pro}:miR172* plants grown under LD. (**K-L**) Box plots showing days to heading (**K**; n ≥ 11) and number of leaves produced by the main tiller (**L**; n ≥ 12). (**M**) Box plots showing *FT1* expression levels determined by qRT-PCR in the 5th (L5) and 7th (L7) leaves. *ACTIN* was used as internal reference. Data correspond to five independent biological replicates. Different letters above the box plots indicate significant differences based on Tukey tests ($P < 0.05$) in panels (**A-D**), (**F**) and (**I**), on Kruskal-Wallis non-parametric testes in (**G**) and (**L**), and on *t*-tests in panels (**H**), (**K**) and (**M**) (Data B in S1 Data). ns = not significant, * = $P < 0.05$, ** = $P < 0.01$, *** = $P < 0.001$. The color of the letters in A-D correspond to the color of the genotypes.

amplify *FT1* transcripts in wild type or *UBI_{pro}:miR172* first leaf samples, but we detected a low *FT1* expression level in *vrn2* mutants (Fig 2H). Interestingly, *UBI_{pro}:miR172 vrn2* showed a significant increase in *FT1* expression compared to the *vrn2* mutant (Fig 2H). *VRN1* expression was detected in all lines and no difference was observed between wild type, *UBI_{pro}:miR172* and *vrn2* samples. However, *VRN1* expression was significantly higher in *UBI_{pro}:miR172 vrn2* plants (Fig 2I). These results indicate that overexpression of miR172 can promote *FT1* and *VRN1* expression in the absence of *VRN2*.

Since *FT1* and *VRN1* can also regulate each other in a positive feedback loop [9,17,42,50,86], we also introduced the *UBI_pro*:*miR172* transgene into a *vrn1 vrn2* mutant background [41] to further dissect these interactions. Plants with loss-of-function mutations in *VRN1* (*vrn1*) cannot repress *VRN2* expression and have very delayed heading [41] (Fig 2J). Elimination of *VRN2* in the *vrn1 vrn2* mutant background accelerates the flowering transition [41] (Fig 2J). Interestingly, the *UBI_pro*:*miR172 vrn1 vrn2* plants headed on average 8 days earlier than *vrn1 vrn2* and produced one fewer leaf (Fig 2J–2L), indicating that *UBI_pro*:*miR172* can promote flowering in the absence of *VRN1* and *VRN2*. We collected tissue from the fifth and seventh leaves of these plants and measured *FT1* mRNA levels. In agreement with the rapid flowering, we observed a significant increase in *FT1* expression in seventh leaves of *UBI_pro*:*miR172 vrn1 vrn2* compared to *vrn1 vrn2* plants (Fig 2M). Taken together, these results indicate that miR172 can promote flowering through the regulation of *FT1* expression in leaves independently of *VRN1* and *VRN2*.

## miR156 modulates wheat flowering time through regulation of miR172 and *AP2L1* expression

Next, we examined the connection between the miR172-*AP2L* module described above and the developmentally regulated miR156. A time-course experiment in leaves of wild type Kronos grown under LD conditions showed higher levels of miR156 in the first and third leaves followed by downregulation in adult fifth and seventh leaves (Fig 3A). In the same samples, mature miR172 levels increased with plant age, following an opposite trend to miR156 (Fig 3B). These patterns are similar to those observed in other species [76,78,83], suggesting that the age-related miR156-miR172 pathway is conserved in wheat.

To investigate the role of miR156 on the regulation of the miR172-*AP2L* module in wheat, we first generated transgenic plants expressing constitutively high levels of miR156. We identified 5 loci per genome in tetraploid wheat that encode miR156 precursor sequences that can be potentially processed to generate mature miR156 sequences (S2 Table). We decided to clone and overexpress the miR156b,c locus, which has been studied in several monocot species [87]. Interestingly, the miR156b,c loci from rice, maize and *Brachypodium* include two precursors that are expressed in tandem, while *Triticeae* species contain an extra precursor so the wheat miR156b,c locus includes three precursors in tandem (S6A Fig).

Overexpression of the B genome miR156b,c locus in tetraploid Kronos using the maize *UBIQUITIN* promoter resulted in $T_0$ plants that produced many tillers and had a bushy appearance (S6B Fig), as previously reported [88]. However, these plants were sterile and did not set grain. To generate stable transgenic lines overexpressing miR156, we used the LhG4/pOp two-component system ([89]; see details in Material and Methods). In this system, miR156 was only overexpressed after crossing plants harboring *UBI_pro*:*LhG4* with those containing the *pOp*:*miR156* construct (Fig 3C). Compared with $F_1$ plants harboring only one of the constructs, the $F_1$ *UBI_pro*:*LhG4*/*pOp*:*miR156* plants expressed high levels of miR156 (Fig 3D), required 10 more days to head (Fig 3E) and produced on average 8 more leaves than the controls (Fig 3F). The delayed flowering of the *UBI_pro*:*LhG4*/*pOp*:*miR156* plants correlated with a significant reduction in *FT1*, *FUL2*, and *FUL3* expression (Fig 3G–3I), a significant reduction in mature miR172 expression (Fig 3L), and higher levels of *AP2L1* (Fig 3M) compared with the controls. The expression levels of *VRN1*, *VRN2* and *AP2L5* were not affected by the overexpression of miR156 (Fig 3J, 3K, and 3N).

To complement the analysis of the transgenic lines overexpressing miR156, we generated transgenic lines expressing a target mimic to reduce miR156 activity *in vivo* (MIM156) (Fig 4A). We generated four independent MIM156 lines in Kronos that had lower levels of miR156

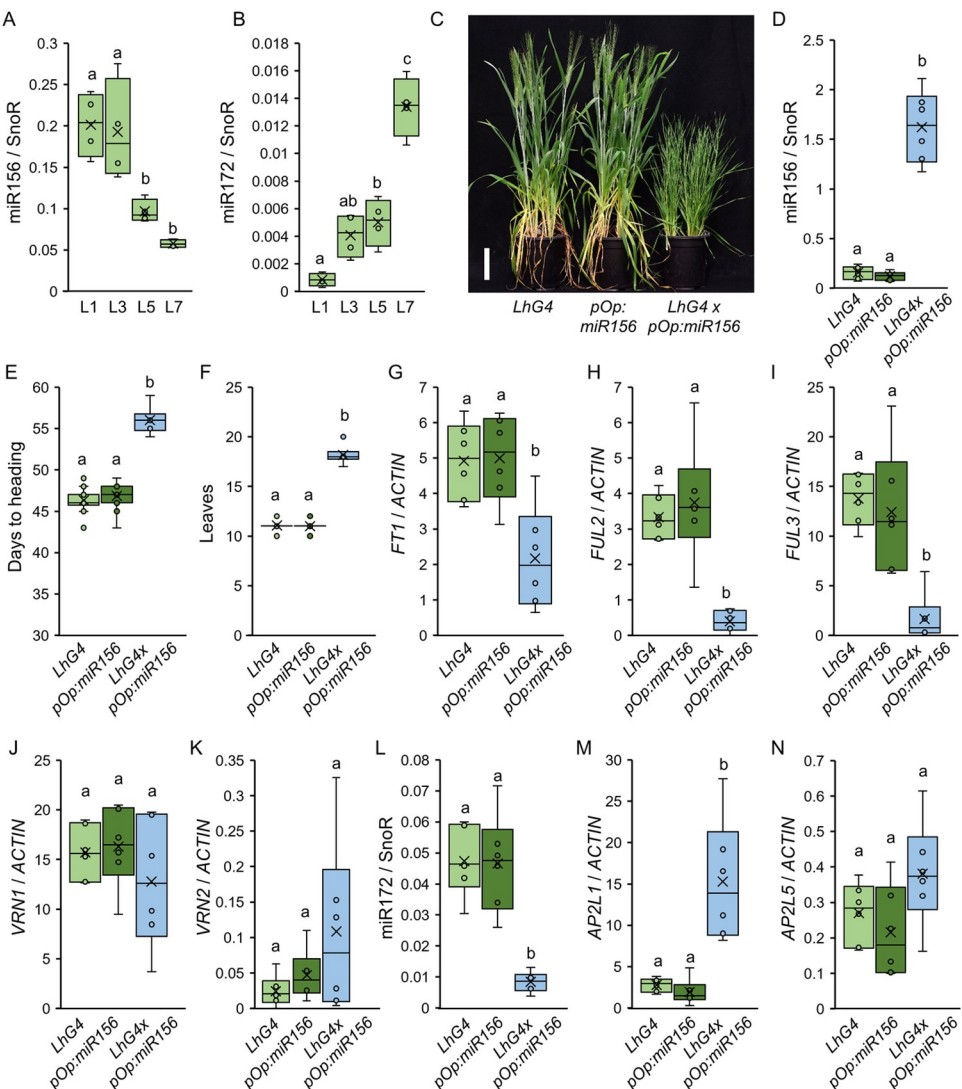

**Fig 3. Overexpression of miR156 delays the flowering transition in wheat.** (**A-B**) Box plots showing the expression levels of miR156 (**A**) and miR172 (**B**) determined by qRT-PCR in the 1st (L1), 3rd (L3), 5th (L5) and 7th (L7) leaves of wild type Kronos plants growing under LD. SnoR was used as an internal reference. Data correspond to four independent biological replicates. (**C-N**) *UBI_pro:LhG4* (*LhG4*), *pOp:miR156* and *UBI_pro:LhG4/pOp:miR156* plants growing under LD. (**C**) Six-week-old plants. Scale bar = 10 cm. (**D**) Box plots showing miR156 expression levels determined by qRT-PCR in the 9th leaf. Data correspond to six independent biological replicates. (**E-F**) Box plots showing days to heading (**E**; n ≥ 6) and the number of leaves produced by the main tiller before the transition to the spike (**F**; n ≥ 6). (**G-N**) Box plots showing the expression levels of *FT1* (**G**), *FUL2* (**H**), *FUL3* (**I**), *VRN1* (**J**), *VRN2* (**K**), miR172 (**L**), *AP2L1* (**M**) and *AP2L5* (**N**) determined by qRT-PCR in the 9th leaf. *ACTIN* was used as internal reference for *FT1*, *FUL2*, *FUL3*, *VRN1*, *VRN2*, *AP2L1* and *AP2L5*, and SnoR was used for miR172. Data correspond to six independent biological replicates. Different letters indicate significant differences in Tukey tests ($P < 0.05$), except for panel (**F**) where non-parametric Kruskal-Wallis pair-wise tests were used (Data C in S1 Data).

(Fig 4B) and characterized their flowering phenotypes under LD conditions. The MIM156 plants headed 4 days earlier (Fig 4C) and produced 1–2 fewer leaves than the non-transgenic sister lines (Fig 4D), indicating an early transition to reproductive development. We then checked the expression of flowering genes in the first, third and seventh leaves of wild type and MIM156#2 plants grown under LD conditions. Consistent with their early flowering, the third

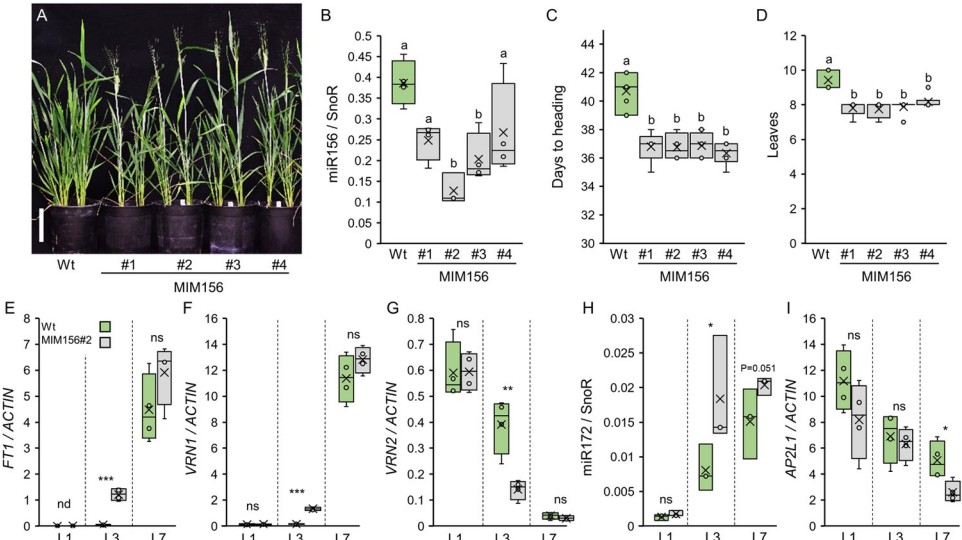

**Fig 4. Transgenic target mimic MIM156 induces expression of flowering promoting genes and accelerates heading time.** (**A**) Five-week-old wild type Kronos (Wt) and $T_1$ plants of four independent MIM156 transgenic lines grown under LD. Scale bar = 10 cm. (**B**) Box plots showing the expression levels of miR156 determined by qRT-PCR in the $1^{st}$ leaves (L1) of wild type Kronos plants (Wt) and $T_1$ plants of four independent MIM156 lines grown under LD. SnoR was used as internal reference. Data correspond to four independent biological replicates. (**C-D**) Box plots showing days to heading (**C**; n ≥ 4) and the number of leaves produced by the main tiller (**D**; n ≥ 4) for wild type Kronos (Wt) and four independent MIM156 $T_1$ lines grown under LD. (**E-I**) Box plots showing the expression levels of *FT1* (**E**), *VRN1* (**F**), *VRN2* (**G**), miR172 (**H**) and *AP2L1* (**I**) determined by qRT-PCR in the $1^{st}$ (L1), $3^{rd}$ (L3) and $7^{th}$ (L7) leaves of wild type Kronos (Wt) and MIM156#2 growing under LD. As internal references, we used *ACTIN* for *FT1*, *VRN1*, *VRN2* and *AP2L1*, and SnoR for miR172. Data correspond to four independent biological replicates. Different letters above the box plots indicate significant differences based on Tukey tests in panels B and D and Kruskal-Wallis tests in panel C ($P < 0.05$). Differences in panels E-I are based on *t*-tests (Data D in S1 Data). ns = not significant, $^* = P ≤ 0.05$, $^{**} = P ≤ 0.01$, $^{***} P ≤ 0.001$.

leaves of MIM156#2 plants expressed higher levels of *FT1* (Fig 4E) and *VRN1* (Fig 4F), and lower levels of *VRN2* (Fig 4G).

We hypothesized that part of the rapid flowering phenotype of MIM156 plants was mediated by the induction of miR172. In agreement with this hypothesis, we observed higher levels of miR172 in the third and seventh leaves of MIM156 plants (Fig 4H), and a reduction in *AP2L1* expression in the seventh leaf (Fig 4I). In addition, we crossed MIM156 and MIM172 transgenic plants and analyzed the flowering phenotype of the $F_2$ population. As expected, the MIM156 MIM172 plants were later and produced more leaves than single MIM156 plants (S7A and S7B Fig and Data L in S1 Data). However, MIM156 could still accelerate flowering in the presence of MIM172 when compared to single MIM172 (S7A and S7B Fig).

Taken together, these results show that miR156 delays flowering in wheat, in part through the regulation of miR172 and *AP2L1* expression.

## Repression of *AP2L1* is decoupled from the age-controlled downregulation of miR156 in winter wheat plants

The previous results showed that miR156 levels affect the expression of miR172 and *AP2L1* in leaves, which in turn modulates the flowering transition in spring wheat cultivars. However, in spring cultivars the flowering transition occurs rapidly, and it is difficult to distinguish the individual effects of plant age (miR156) and flowering induction on the miR172-*AP2Ls* module. Therefore, we studied the miR156-miR172 pathway during the flowering transition in

winter wheat plants, which require a long exposure to cold temperatures in order to cause rapid flowering. To this end, we used a Kronos TILLING line that has a loss-of-function mutation in the spring *VRN-A1* allele and a functional winter *vrn-B1* allele, and therefore these plants have a winter growth habit ([41], henceforth "winter Kronos"). These lines show accelerated heading time and reduced leaf number with increased vernalization treatments, a response that is saturated after 6 weeks of vernalization (S8A and S8B Fig and Data M in S1 Data).

First, we performed a detailed time-course expression analysis in leaves collected from spring and winter Kronos plants of different ages grown under LD in the absence of cold. Under this condition, winter plants required ~100 days more than spring sister lines to head (S9A and S9B Fig and Data N in S1 Data). In spring plants, the expression of *VRN2* was significantly downregulated while *FT1* and *VRN-B1* transcript levels were induced in the seventh leaf (L7; W5 = 5-week-old plants) compared to the first leaf (L1; W1 = 1-week-old plants), in agreement with their early flowering (Fig 5A–5C). In contrast, the expression of *VRN2* remained high in 5 to 10-week-old winter plants, and then slowly declined until reaching very low levels in 18-week-old plants, which had already developed the flag leaf (Fig 5A). The downregulation of *VRN2* was accompanied by an induction of *FT1* and *VRN-B1* between 14 and 18 weeks (Fig 5B and 5C).

We observed that mature miR156 was expressed at similar levels in juvenile leaves of spring and winter plants (W1), and its expression was then repressed to similar levels in the seventh leaves (W5) of plants with both growth habits (Fig 5D). Interestingly, the miR156 expression continued to show a gradual downregulation in later leaves of winter plants.

Between W1 and W5, miR172 levels increased in both spring and winter Kronos, but reached significantly higher levels in the spring lines (which were close to heading) than in the winter lines (Fig 5E). By contrast, the expression level of *AP2L1* was significantly higher in the seventh leaves of winter plants compared with spring plants at W5 (Fig 5F).

In winter lines, we observed a progressive increase in the expression of miR172 with age until week 16 (Fig 5E). Interestingly, we did not observe a parallel downregulation of the transcript levels of *AP2L1* (Fig 5F) and *AP2L5* (S9C Fig) after W5. This result is similar to the limited effect of $UBI_{pro}$:*miR172* on *AP2Ls* transcript levels described in S2A and S2B Fig. After W14, we observed an abrupt downregulation of *AP2L1* and *AP2L5* that coincided with the upregulation of *FT1* (Fig 5B), and the *SQUAMOSA* genes *VRN1* (Fig 5C), *FUL2* and *FUL3* (S9D and S9E Fig).

For all the genes in Figs 5A–5F (Data E in S1 Data) and S9D and S9E Fig (Data N in S1 Data), the ANOVAS for the differences in gene expression along development were significant for both spring and winter lines, confirming that all these genes are developmentally regulated. Only *AP2L5* (S9C Fig) showed no significant differences for the spring line and only marginally significant differences for the winter lines ($P = 0.05$). In addition, no significant differences in gene expression were detected between the spring line at W5 and the winter line at W18 in Figs 5A–5F (Data E in S1 Data) and S9D-E (Data N in S1 Data), suggesting a similar regulation of these genes at heading time in the two genotypes.

The downregulation of *AP2L1* associated with the upregulation of *SQUAMOSA MADS-box* genes correlates with the negative regulation of *AP2L* genes by *SQUAMOSA MADS-box* genes reported in Arabidopsis [70,90]. A similar regulation may also explain the rapid repression of *AP2L1* (and induction of miR172) observed at W5 in the spring Kronos lines. To further explore this hypothesis, we studied the expression of *AP2L1* and miR172 in vernalized winter Kronos and *vrn1*-null mutant plants [41]. We vernalized 2-week-old winter plants for 8 weeks and collected L7 samples from non-vernalized and vernalized winter plants two weeks after the plants were taken out of the cold. The expression of *VRN2* remained repressed in the

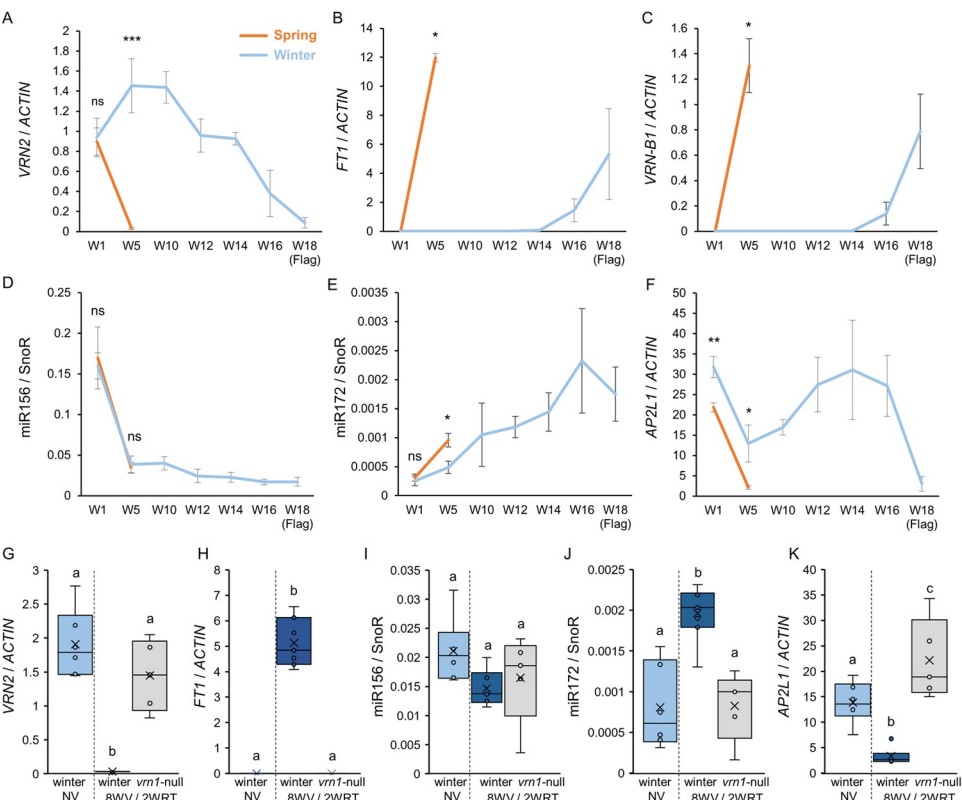

**Fig 5. Effect of vernalization and *VRN1* expression on the transcript levels of flowering genes, miR156 and miR172 in spring and winter wheat. (A-F)** Time-course expression levels determined by qRT-PCR in leaves of spring (orange) and winter (blue) Kronos plants grown under LD conditions in the absence of a vernalization treatment. W1 = week 1, 1st leaf; W5 = week 5, 7th leaf; W10 = week 10, leaves 10th-11th; W12 = week 12, leaves 13th-14th; W14 = week 14, leaves 15th-16th; W16 = week 16, leaves 18th-19th; W18, week 18, flag leaves (20th-23rd). In both genotypes the fully expanded 7th leaves were collected during the 5th week. However, leaf samples were not collected on the same day (34 days for spring and 36–38 days after planting for winter). *ACTIN* was used as internal reference for (**A**) *VRN2*, (**B**) *FT1*, (**C**) *VRN1*, and (**F**) *AP2L1*, and SnoR for (**D**) miR156 and (**E**) miR172. * = $P < 0.05$, ** = $P < 0.01$, and *** = $P < 0.001$ in *t*-tests (A, D-F) and non-parametric Kruskal-Wallis tests (B and C) comparing expression at the same leaf in spring and winter Kronos plants. (**G-K**) Box plots showing the expression levels of *VRN2* (**G**), *FT1* (**H**), miR156 (**I**), miR172 (**J**) and *AP2L1* (**K**) determined by qRT-PCR in the 7th leaf of winter Kronos plants without vernalization, and the 7th leaves of winter Kronos and *vrn1*-null mutants that were vernalized for 8 weeks and then moved to room temperature for two weeks. Samples from the 7th leaf were collected when the leaves were fully expanded. *ACTIN* was used as internal references for *VRN2*, *FT1* and *AP2L1*, and SnoR for miR156 and miR172. Data correspond to at least 5 independent biological replicates. Different letters above the box plots in panels G-K indicate significant differences ($P < 0.05$) in Tukey tests, except for pane (**H**), where a non-parametric Kruskal-Wallis test was used (Data E in S1 Data).

vernalized winter plants 2 weeks after returning to room temperature but not in the *vrn1*-null mutant, consistent with previous findings [41] (Fig 5G). Expression of *FT1* (Fig 5H), *VRN-B1*, *FUL2* and *FUL3* (S9G–S9I Fig) was induced only in vernalized winter plants with the functional *VRN1* allele. The expression of miR156 was not modified by the vernalization treatment (Fig 5I), but we observed an induction of miR172 expression in the vernalized winter plants compared with non-vernalized winter and vernalized *vrn1*-null plants (Fig 5J). Moreover, the expression of *AP2L1* was significantly downregulated in winter plants compared with the *vrn1*-null mutants after both lines were returned to room temperature from the vernalization treatment (Fig 5K).

Taken together, these results indicate that in the leaves of winter plants, the downregulation of miR156 expression with plant age is decoupled from the downregulation of *AP2L* genes

expression, and that the induction of *VRN1* by vernalization promotes the upregulation of miR172 and the repression of *AP2L1*.

## miR172 modulates flowering time in winter wheat plants exposed to different vernalization treatments

We next explored the role that the miR172-*AP2L* module has in the flowering transition of winter wheat plants. To this end, we transferred the *UBI*$_{pro}$:*miR172* and MIM172 transgenes into the winter Kronos background, and evaluated the flowering time of plants grown under LD conditions without vernalization, or of those vernalized for 2, 3 and 6 weeks once the plants reached the second leaf stage. Planting times were coordinated to ensure that both vernalized and non-vernalized plants reached the same number of leaves by the end of the different vernalization treatments. Days to heading were measured from the time the vernalized plants were removed from the cold treatment (Figs 6 and 7).

In the absence of vernalization, winter plants took on average 146.5 days to head from the time all other plants were taken out of the cold and produced 25 leaves on the main tiller (Fig 6A and 6B). Under this condition, winter MIM172 and *vrn1*-null headed 18 days later than the winter plants, whereas winter *UBI*$_{pro}$:*miR172* lines headed 46 days earlier than the winter plants and produced 18 leaves (Fig 6A and 6B). Days to heading and leaf number showed similar profiles across genotypes suggesting that the differences in heading time were due to developmental differences in the timing of the transition between the vegetative and reproductive phase and not simply due to differences in the duration of spike development and stem elongation after the transition. After 6 weeks of vernalization, which is an almost saturating vernalization treatment for winter Kronos (S8A and S8B Fig), winter *UBI*$_{pro}$:*miR172*, winter MIM172, and winter Kronos plants headed rapidly (33 to 39 days after plants were removed from the cold) and showed similar leaf numbers (9 to 9.5 leaves), with only small differences between lines (Fig 6A and 6B). Under the same 6 weeks of cold treatment, *vrn1*-null plants showed a weaker response to vernalization, headed after 136 days, and produced 24.3 leaves, consistent with previous findings [41].

A two-week sub-saturating vernalization treatment accelerated the reproductive transition, with winter Kronos heading 125 days after the cold treatment and producing 21.7 leaves and winter *UBI*$_{pro}$:*miR172* lines heading 81.5 days after the cold treatment and producing 15.3 leaves (Fig 6A and 6B). Compared to non-vernalized plants, the two-week vernalization treatment produced a stronger acceleration of heading time and reduction in leaf number in winter Kronos (-21.5 days and -3.83 leaves) and *UBI*$_{pro}$:*miR172* (-19.3 days and -3.3 leaves) than in MIM172 (-1.7 days and -1.2 leaves). Similar trends were observed when plants were exposed to a three-week sub-saturating vernalization treatment (Fig 6A and 6B). Interestingly, the MIM172 winter plants, which showed a weaker response, headed at a similar time as *vrn1*-null plants under both sub-saturating vernalization treatments. The flowering differences were reflected in a highly significant genotype x vernalization interaction in factorial ANOVAs for both traits ( $P < 0.001$, Data F in S1 Data). Even when we removed the *vrn1*-null genotype from the analysis, this interaction remained highly significant ( $P < 0.001$ Data F in S1 Data). This result confirmed that miR172 modulates the flowering response under sub-saturating vernalization treatments in winter wheat.

To further characterize the interaction between vernalization and miR172, we collected the ninth leaf 25 days after the plants were taken out of the cold and performed gene expression analyses. Comparison of miR172 levels in the leaves of winter Kronos and *vrn1*-null plants vernalized for 6 weeks showed a slight increase in miR172 expression in the plants with the functional *VRN1* allele (Fig 6C, Data F in S1 Data). Although this increase in miR172 expression is

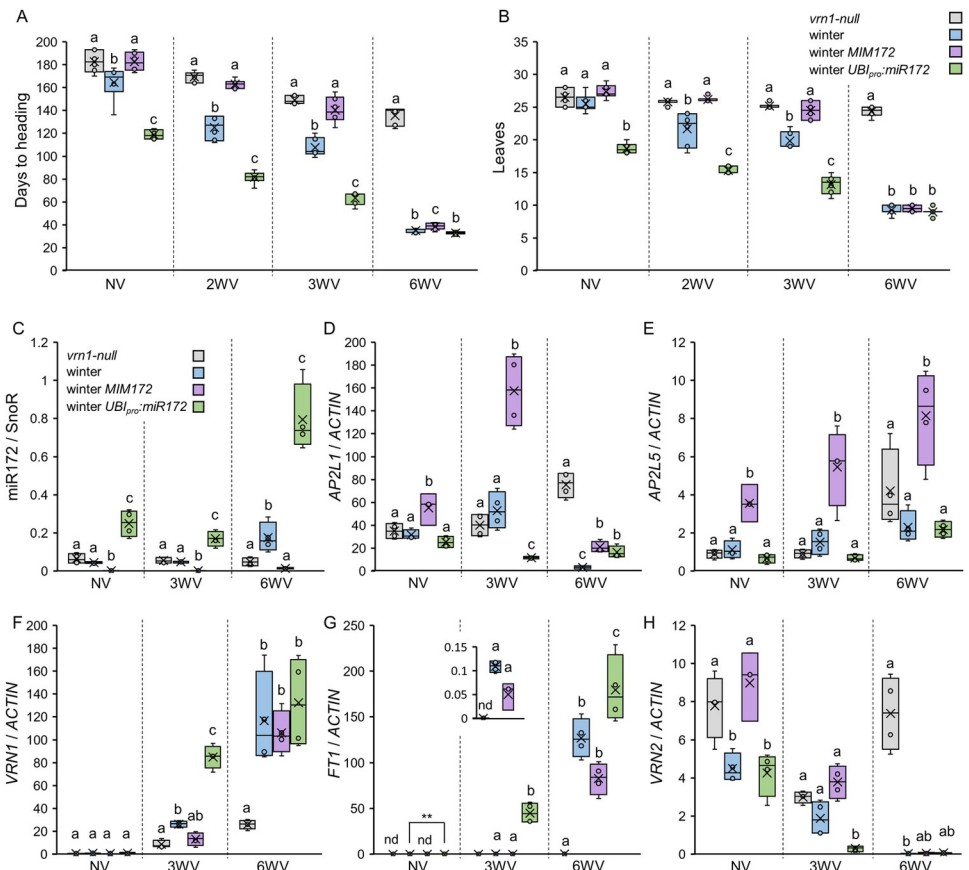

**Fig 6. miR172 levels modulate the flowering response in winter wheat plants.** (**A-B**) Box plots showing days to heading calculated from the end of vernalization treatments (**A**) or number of leaves in the main tiller (**B**) for *vrn1*-null, winter Kronos, winter MIM172, and winter *UBI_{pro}:miR172* plants grown under LD conditions without vernalization (NV), or with 2 (2WV), 3 (3WV) or 6 (6WV) weeks of vernalization (n ≥ 6). (**C-H**) Expression levels of miR172 (**C**), *AP2L1* (**D**), *AP2L5* (**E**), *VRN1* (**F**), *FT1* (**G**) and *VRN2* (**H**) determined by qRT-PCR in the 9th leaf of *vrn1*-null, winter Kronos, winter MIM172, and winter *UBI_{pro}:miR172* plants grown under LD conditions in the absence of a vernalization treatment, and after 3 (3WV) and 6 (6WV) weeks of vernalization. After vernalization, the plants were moved to room temperature for 25 days before the fully expanded 9th leaf was collected. SnoR was used as internal reference for miR172, and *ACTIN* was used as internal reference for the other genes. Data correspond to four independent biological replicates. Different letters above the box-plots indicate significant differences ($P$ <0.05) in Tukey tests performed within each vernalization treatment, except for panel (**B**) 2WV where a non-parametric Kruskal-Wallis test was used (Data F in S1 Data).

consistent with the previous result after 8 weeks of vernalization (Fig 5J), the difference between the two lines was not significant after 6 weeks vernalization. Moreover, the partial vernalization treatment of 3 weeks was not enough to induce miR172 levels, which showed similar levels in leaves of both winter Kronos and *vrn1*-null plants (Fig 6C).

The ectopic expression of *UBI_{pro}:miR172* in winter plants resulted in significantly higher levels of miR172 in all three vernalization treatments, with the highest levels observed after 6 weeks of vernalization (Fig 6C). By contrast, MIM172 lines showed significantly lower levels of miR172 expression in all three vernalization treatments, which were inversely correlated with changes in *AP2L5* expression (Fig 6E). *AP2L1* transcript levels were more than an order of magnitude higher than those of *AP2L5*, and were not significantly different among genotypes (Fig 6D).

We also quantified the expression of the flowering genes *VRN1*, *FT1*, and *VRN2* in the same samples (Fig 6F–6H). Both *VRN1* and *FT1* showed a gradual increase in expression that correlated with the duration of the vernalization treatment, with the highest values observed after 6 weeks of cold (Fig 6F and 6G). Interestingly, after 3 weeks of vernalization, winter *UBI~pro~:miR172* plants expressed higher levels of *FT1* and *VRN1*, but the differences were significant only for *FT1* (relative to *vrn1*-null and MIM172). After 6 weeks of vernalization, winter Kronos, winter *UBI~pro~:miR172* and winter MIM172 plants expressed similar levels of *VRN1* and *FT1* that were significantly higher than in the *vrn1*-null plants (Fig 6F and 6G). An opposite trend was observed for *VRN2*, where expression levels decreased as the time in the cold increased, except in *vrn1*-null plants, which showed high *VRN2* expression even after 6 weeks of vernalization (Fig 6H). As expected, higher *VRN1* transcript levels in winter Kronos, *UBI~pro~:miR172* and MIM172 plants after 6 weeks of vernalization were associated with lower *VRN2* mRNA levels (Fig 6H).

## *AP2L1* and *AP2L5* play significant roles in the modulation of flowering time in winter wheat exposed to sub-saturating vernalization treatments

To test if the effects of miR172 on the flowering response to vernalization in winter wheat were mediated by its repression of the *AP2L* genes, we crossed *ap2l5* and *ap2l1* mutations with winter Kronos lines and assessed heading time without vernalization or with a sub-saturating 2-week vernalization treatment (Fig 7). Under both conditions, both single *ap2l5* and *ap2l1* mutants headed 39 to 57 days earlier and had 5.5 to 7.7 fewer leaves than the winter plants and

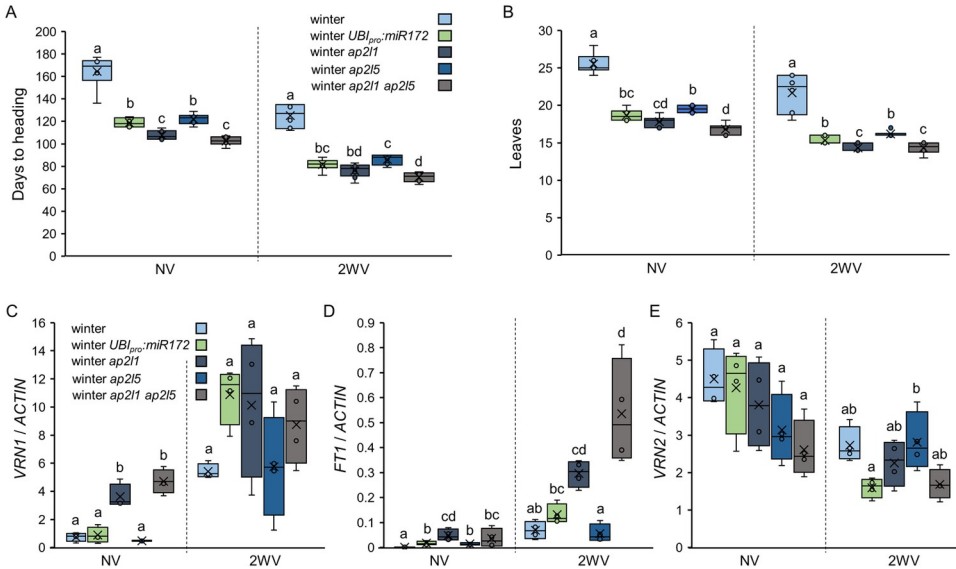

**Fig 7. *AP2L1* and *AP2L5* genes modulate the flowering response in winter plants.** (**A-E**) Winter Kronos, winter *UBI~pro~:miR172*, winter *ap2l1*, winter *ap2l5* and winter *ap2l1 ap2l5* grown under LD conditions without vernalization (NV) and with 2 (2WV) weeks of vernalization. Box plots showing (**A**) days to heading calculated from the end of vernalization treatments or (**B**) number of leaves in the main tiller (n ≥ 6). (**C-E**) Expression levels of *VRN1* (**C**), *FT1* (**D**) and *VRN2* (**E**) determined by qRT-PCR in the 9th leaves. After vernalization, plants were moved to room temperature for 25 days before the fully expanded 9th leaf was collected. *ACTIN* was used as internal reference. (**F-H**) In a second experiment with 2 weeks of vernalization, samples were collected from leaves 12th and 15th to determine expression levels of (**F**) *VRN1*, (**G**) *FT1* and (**H**) *VRN2*. Data correspond to four independent biological replicates (except for the 15th leaf, where six replicates were used). Different letters above the box plots indicate significant differences ($P < 0.05$) in Tukey tests, except for panel (**B**) 2WV where non-parametric Kruskal-Wallis tests were used (Data G in S1 Data).

were similar to winter *UBI_{pro}:miR172* transgenic plants (Fig 7A and 7B). The *ap2l1* mutant was significantly earlier (10–14 days) and had less leaves (1.7–1.8 leaves) than the *ap2l5* mutant in both treatments (Fig 7A and 7B).

The winter *ap2l1 ap2l5* combined mutant showed the earliest heading and fewest number of leaves relative to all other genotypes in the absence of vernalization (102 days and 16.8 leaves) or when partially vernalized (70 days and 14.3 leaves), and in both cases it headed earlier and had fewer leaves than winter *UBI_{pro}:miR172* plants (Fig 7A and 7B). It is worth noting that the non-vernalized *ap2l1 ap2l5* mutant still headed one month later than fully vernalized winter Kronos plants or non-vernalized spring Kronos, indicating that the *ap2l1 ap2l5* mutant reduced but did not abolish the vernalization requirement in wheat.

In plants exposed to 2 weeks of vernalization, the transcript levels of *FT1* and *VRN-B1* in the 9$^{th}$ leaf (Fig 7C, 7D) were very low, and no significant differences among genotypes were detected. Therefore, we performed an additional 2-weeks vernalization experiment, collecting sample from the 12$^{th}$ and 15$^{th}$ leaves (Fig F-H). The differences in *VRN-B1* and *FT1* transcript levels among genotypes were not significant in the 12$^{th}$ leaf but were significant in the 15$^{th}$ leaf. Consistent with their earlier heading time, the *ap2l1 ap2l5* mutant and the *UBI_{pro}:miR172* transgenic plants showed significantly higher levels of *VRN1* and *FT1* and significantly lower levels of *VRN2* than the winter control (Fig 7F-7H).

Finally, we combined the MIM172 and *UBI_{pro}:AP2L-B1* transgenes in the winter Kronos background. We expected both MIM172 and *UBI_{pro}:AP2L-B1* to delay flowering in response to vernalization treatments; thus, we analyzed the progeny under a 4-week partial vernalization experiment. Heading of the MIM172 plants was delayed relative to the winter Kronos line, but both MIM172 and winter Kronos were earlier than the *vrn1*-null plants (S10 Fig and Data O in S1 Data). Interestingly, winter plants harboring both MIM172 and *UBI_{pro}:AP2L-B1* were later than MIM172 plants and headed at a similar time as the *vrn1*-null plants (S10 Fig).

Taken together, these results show that in winter wheat cultivars, the levels of miR172 and *AP2L* genes affect the duration of cold exposure required to induce changes in the expression of *VRN1*, *VRN2*, and *FT1* genes, which in turn can modulate the flowering response to sub-saturating vernalization conditions.

## Discussion

### *AP2L* genes function as flowering repressors

In this work, we used a combination of genetic, transgenic, and expression analyses to demonstrate that the miR172-*AP2L* module plays important roles in the regulation of flowering time in wheat through interaction with central players of the vernalization pathway.

AP2L proteins are transcription factors that function as repressors of the flowering transition in several species, including eudicot species such as Arabidopsis and monocots such as rice [54]. In these species, the *AP2L* gene family includes several members that usually have overlapping roles in the regulation of the flowering transition. Arabidopsis for example contains six *AP2L* genes (*TOE1*, *TOE2*, *TOE3*, *SCHLAFMUTZE (SMZ)*, *SCHNARCHZAPFEN (SNZ)*, and *AP2*), and a previous study showed that the *toe1 toe2* double mutants flowered earlier than either single mutant, but still it was not as rapid flowering as the *35S_{pro}:miR172* over-expressing lines [58]. A more recent study showed that the *toe1 toe2 smz snz toe3*-1 *ap2*-12 sextuple mutant phenocopied the *35S_{pro}:miR172* plants very closely [55].

Our work also showed overlapping roles for *AP2L1* and *AP2L5* genes in the control of the flowering transition in wheat, with both single and combined *ap2l1 ap2l5* mutants flowering earlier than the sister wild type lines in spring and winter wheat. The *ap2l1 ap2l5* mutant was earlier than the single mutants and the *UBI_{pro}:miR172* transgenic plants. The relatively mild

effect of $UBI_{pro}$:*miR172* on heading time can be an indirect effect of our selection of a fertile transgenic line with a weak floret phenotype, which likely expresses intermediate levels of the transgene. In our previous study, strong overexpression of miR172 in Kronos resulted in floret defects and sterile plants that could not be propagated [84].

In addition to its effect on flowering time, *AP2L5* affected inflorescence and flower development, and the *ap2l5* mutant defects were magnified when combined with *ap2l2* mutants [84]. However, *ap2l1* mutants developed normal inflorescences (S2 Fig). Similar observations were described in Arabidopsis, with *TOE1* (ortholog of *AP2L1*) and *TOE2* affecting flowering time, and *AP2* (closer to *AP2L2* [84]) regulating both flowering time and flower development [55,58,72]. These results suggest an ancient sub-functionalization of these two *AP2L* clades that is still maintained in Arabidopsis and wheat.

The mechanisms by which AP2L transcription factors control flowering are best described in Arabidopsis. Chromatin immunoprecipitation analyses revealed that TOE1 and SMZ proteins can bind to *FT* promoter sequences and inhibit its expression in leaves [56,57]. In addition, these genome-wide experiments showed that SMZ and AP2 repress many other flowering time regulators acting downstream of *FT* in the leaves and the SAM [55,56]. Besides this direct transcriptional regulation of *FT* and other flowering genes, TOE proteins were shown to inhibit CO activity in Arabidopsis [57], as they can interact with the transcriptional activation domain of CO and affect CO protein stability. Similarly, our results indicate that miR172 can induce *FT1* expression in the leaves, likely through the repression of *AP2L* genes, and that this mechanism can occur in the absence of *VRN1* and *VRN2*, or changes in the expression of *PPD1*. However, we currently do not know if this transcriptional regulation of *FT1* results from an indirect effect or is mediated by direct binding of AP2L proteins to *FT1* regulatory sequences.

## Variable contributions of *AP2L* genes to flowering in spring and winter wheats under different photoperiods

*VRN2* and *AP2L* genes are repressors of flowering that likely act by repressing *FT1* transcription in the leaves. However, the elimination of both types of *FT1*-repressing genes in $UBI_{pro}$: *miR172 vrn2 vrn1* plants did not accelerate flowering to the same extent observed in backgrounds harboring the spring *Vrn1* allele [$UBI_{pro}$:*miR172 vrn2 vrn1*: ~55 days (Fig 2K) *vs.* $UBI_{pro}$:*miR172 vrn2 Vrn1*: ~35 days (Fig 2F)]. This result indicates that *VRN1* can accelerate flowering independently of *VRN2* and *AP2L* genes. This could be mediated by a direct regulation of *FT1* in the leaves, a hypothesis supported by *VRN1* promotion of *FT1* expression in leaves of *vrn2*-null plants [42], and by the binding of VRN1 to the *FT1* promoter in a CHIP-seq experiment in barley [86]. In addition, *VRN1* was shown to promote the transition of the SAM between the vegetative and early reproductive stages even in the absence of *FT1*, but *FT1* expression was still necessary at later stages for normal spike development and stem elongation [91,92].

The relative contributions of *VRN2* and *AP2L* genes to the repression of flowering in wheat depend on the genetic background. In spring cultivars, both pathways contribute similarly to the repression of flowering, and $UBI_{pro}$:*miR172 vrn2* plants showed an additive effect with very rapid flowering (Fig 2). However, in winter cultivars grown in the absence of vernalization, *VRN2* plays a more predominant role, since *vrn2 vrn1* plants headed within 65 days (Fig 2K), while $UBI_{pro}$:*miR172 vrn1* required 119 days from germination to heading (Fig 6A, 101 days when calculated from the end of the cold treatment). The prominent role of *VRN2* is also supported by the observation that, in both diploid wheat and barley, major QTLs for spring growth habit have been mapped to multiple loss-of function alleles of *VRN2* [29,35]. Our

results indicate that although miR172 and *AP2L* genes play a more limited role than *VRN2* in the repression of flowering in non-vernalized winter plants, they can modulate the flowering response under non-saturating vernalization conditions.

Interactions between *AP2L* genes and vernalization have also been described in the perennial brassica species *A. alpina*. In this species, the orthologs of Arabidopsis *AP2* [60,61] and *TOE2* [93] regulate the age-dependent response to vernalization and contribute to the perennial growth habit. However, whereas loss-of-function mutations in *A. alpina* ortholog of *AP2* were sufficient to confer a spring growth habit [60,61], winter wheat plant overexpressing *UBI-pro*:*miR172* or carrying *ap2l1 ap2l5* mutations showed a reduced but still strong vernalization requirement (Figs 6A, 6B, 7A, and 7B). These observations confirm a conserved role of *AP2L* genes as flowering repressors and indicate that their relative contribution to the vernalization requirement can vary among species.

Our results also show that the miR172-*AP2L* module regulates Kronos flowering time under SD, and that the effects were stronger under SD (12 d) than under LD (4 d, Figs 1 and S1). In maize, where some varieties behave as SD plants, a major flowering-time QTL, named *Vegetative to generative transition 1* (*Vgt1*), was mapped to a 2-kb conserved noncoding region positioned 70 kb upstream of *ZmTOE1* [63]. *Vgt1* acts as a cis-acting regulatory element that affects the transcript expression levels of *ZmTOE1* and flowering time in maize. Interestingly, a recently identified short-day flowering time response QTL on wheat chromosome 1BS has also been linked to *AP2L-B1* (*TaTOE1*) [94]. Since the *AP2L-B1* mutant alleles were associated with earlier flowering, the authors suggested that *AP2L-B1* likely works as a flowering repressor, which agrees with the results presented here.

These natural *AP2L1* alleles, together with the mutant alleles generated in this work, provide wheat breeders with additional tools to fine-tune wheat flowering time to specific environments.

## Vernalization and plant age control miR172 and *AP2L* expression in leaves

miR156-*SPLs* and miR172-*AP2Ls* are two conserved miRNA modules in plants that display a complementary expression pattern in the shoot with plant age [78,82,83]. Our results showed similar complementary expression patterns for miR156 and miR172 in spring wheat. In MIM156 plants with reduced miR156 activity, miR172 expression is higher in juvenile leaves (Fig 4), while adult leaves of plants overexpressing miR156 have lower levels of miR172 and higher levels of *AP2L1* (Fig 3). These results indicate that miR156 is necessary for the correct expression pattern of miR172 and *AP2L1*. However, these interactions are altered in winter wheat backgrounds. While the expression profile of miR156 was similar in spring and winter cultivars, the expression of miR172 differed between these two backgrounds. In spring plants, miR172 expression increased rapidly reaching high levels at early stages following the flowering transition, whereas this increase with age was delayed in winter wheat compared with spring wheat. These results suggest that miR156 and miR172-*AP2Ls* modules are connected, but also that other mechanisms may regulate the expression profile of miR172 downstream of miR156.

We speculate that during the induction of the flowering transition several feedback loops act to reinforce changes in gene expression and to produce a robust shift in the developmental program. Vernalizing winter plants for several weeks promotes rapid induction of miR172 and downregulation of *AP2L1* similar to that observed in spring cultivars. Interestingly, similar results were observed in the perennial brassica *A. alpina*. In this species the accumulation of miR156 is reduced in the shoot apex of older plants that acquire competence to flower in long days, but miR172 expression is low. Vernalization is required to induce flowering and to promote an increase in miR172 levels in the shoot apex [61].

Another interesting observation from this work was that in winter plants *AP2L1* expression was not downregulated with plant age even when miR172 expression increased. It has been shown that miR172 levels can reduce AP2L protein levels without affecting transcript levels [58,72]. It was suggested that miR172 might repress the translation of *AP2L* mRNA [58,72], or that feedback loops involving *AP2L* genes controlling their own transcription act to keep transcript levels constant [85]. However, downregulation of *AP2L1* expression by vernalization (Figs 5 and 6) or in the flag leaf of late flowering winter plants grown without vernalization (Fig 5) suggest that other mechanisms may also exist in wheat to repress *AP2L* gene expression.

A recent study in Arabidopsis has shown that FUL directly represses the transcription of several *AP2L* genes such as *SNZ*, *TOE1*, and *AP2* during inflorescence development in parallel to miR172 [90]. An additional Arabidopsis study indicated that FUL and miR172 control *AP2L* gene activity at both transcriptional and post-transcriptional levels to promote the floral transition [70], which could provide a rapid and robust mechanism to deplete *AP2L* genes as flowering proceeds. This seems to be also the case in wheat, where vernalization of winter wheat lines results in the downregulation of *AP2L1*, but only in lines with active *VRN1* genes. The high levels of *AP2L* genes in vernalized *vrn1*-null plants relative to the vernalized plants with a functional *VRN1* allele indicates that *VRN1* plays a critical role in the transcriptional downregulation of *AP2L1*. Since the overexpression of miR172 had no significant effects on the transcriptional regulation of *AP2L1* and *AP2L5* (S3 Fig), we think that the upregulation of miR172 by *VRN1* (Fig 5J) is insufficient to explain the transcriptional downregulation of *AP2L1* by *VRN1* (Fig 5K).

In winter wheat, vernalization results in the induction of the expression of *VRN1* and its closest paralogs *FUL2* and *FUL3* (Figs 6 and S9G–S9I). These three genes are also induced in older winter plants, even in the absence of vernalization (Figs 5, S9D, and S9E), and the timing of their inductions coincides with the downregulation of *AP2L1* and *AP2L5* (Figs 5F and S9C), providing additional support for the role of *VRN1* in the direct or indirect downregulation of *AP2L* genes in the leaves. However, since these genes are also expressed in wheat developing spikes, we cannot rule out alternative mechanisms in these tissues.

We speculate that the downregulation of *AP2Ls* at later developmental stages could reflect an additional role of these genes on internode elongation. Transgenic wheat plants expressing MIM172 have higher *AP2Ls* expression and shorter internodes than the non-transgenic controls [66]. Similarly, a dominant mutation in the miR172 target site of *AP2L2* in barley negatively regulates peduncle elongation during the reproductive transition [95]. Since the downregulation of *AP2Ls* in flag leaves was associated with an increase in *FT1* expression (Fig 5), and *FT1* is required to promote internode elongation in wheat [92], we speculate that *AP2Ls* may control internode elongation by controlling *FT1* expression in the leaves.

## miR156 regulates flowering by controlling miR172 and other genes

In Arabidopsis, miR156 negatively regulates the expression of multiple *SPL* genes, which directly promote the induction of miR172 expression and the initiation of the reproductive phase [65,78,96–98]. In addition, SPL transcription factors can also modulate flowering by miR172-independent pathways. For example, Arabidopsis SPL3, SPL9 and SPL15 promote the expression of *SOC1*, *FUL*, *AP1* and *LFY* in the SAM [96,98,99], and these interactions are relevant to flowering promotion under non-inductive SD conditions.

In wheat, overexpression of miR156 also results in the repression of several *SPL* genes [88], suggesting that the miR156-SPL module is conserved in wheat. The significantly earlier flowering observed in the transgenic wheat plants including both MIM156 and MIM172 relative to

those including only MIM172 (S7 Fig) indicates that miR156 can also regulate flowering time by miR172-independent pathways. Our data indicate that, in addition to miR172, *FUL2* and *FUL3* might participate in the flowering responses downstream of the miR156-*SPL* module. In adult leaves of the late flowering plants overexpressing miR156, *FUL2* and *FUL3* expression is reduced (Fig 3H and 3I), similar to miR172. Moreover, in the leaves of non-vernalized winter plants, *FUL2* and *FUL3* expression is induced a couple of weeks earlier (W12, S9D and S9E Fig) than *FT1* and *VRN-B1* (W14, Fig 5B and 5C).

Overexpression of *FUL2* using the maize *UBIQUITIN* promoter in Kronos resulted in accelerated flowering [100]; thus, it would be interesting to test whether SPL transcription factors promote flowering in non-vernalized winter plants through the induction of *FUL2* and *FUL3* expression.

## Integrating the miR172-*AP2L* regulatory module into the wheat flowering network

Flowering time must occur at an optimal time to maximize reproductive success. This precision is achieved through a complex regulatory network that senses, translates and integrates different signals into the regulation of a few central flowering genes. In wheat, flowering is promoted by long days and vernalization and both pathways converge on the activation of *FT1* in the leaves (Fig 8). We incorporate into this model the conserved miR156-miR172-*AP2L* module, which constitutes another pathway to regulate *FT1* expression in the leaves (Fig 8).

In this model, both VRN2 and AP2Ls repress *FT1* in the leaves to prevent flowering in the fall. In spring cultivars, the levels of *AP2L* genes are down-regulated developmentally by the sequential action of miR156 and miR172, which promotes a rapid flowering transition in

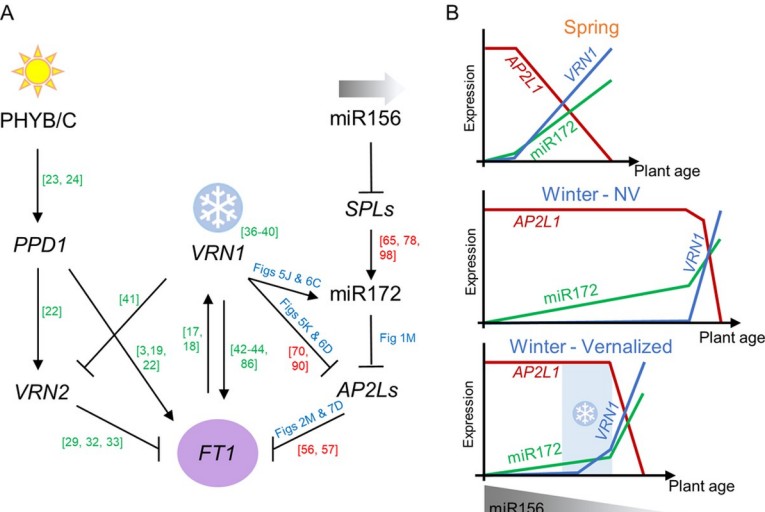

**Fig 8. Working model for flowering regulation in wheat.** (**A**) Flowering network integrating photoperiod, vernalization and age signals into the regulation of *FT1* expression in wheat leaves. Arrows indicate promotion of gene expression and lines ending in a crossed-bar repression. Numbers in brackets indicate the references supporting the different interactions (red, indicates Arabidopsis references). The critical figures supporting the interactions presented in this paper are indicated in blue. (**B**) Model illustrating the expression profile of miR172 and *AP2L1* with plant development. Top panel shows expression profiles of miR172 and *AP2L1* in spring wheat cultivars, where *VRN1* is expressed in the absence of cold. The middle panel shows expression profiles of miR172 and *AP2L1* in a winter wheat background without vernalization (Winter—NV). The bottom panel shows expression profiles in vernalized winter plants (Winter–Vernalized). In this background, *VRN1* expression after vernalization (light blue region) promotes miR172 upregulation and *AP2L1* downregulation. In both spring and winter cultivars, miR156 expression in leaves is downregulated with plant age.

spring cultivars. However, in winter wheat plants, the upregulation of miR172 and the down-regulation of *AP2L* genes requires the induction of *VRN1* during vernalization, which results in an additional regulatory layer that delays the upregulation of *FT1* and prevents premature flowering in the fall. In turn, changes in the dosage or repression activity of AP2Ls affect flowering time in winter wheat backgrounds. Under sub-saturating conditions, plants with lower dosage or activity of AP2L require less *VRN1* and lower reductions in *VRN2* expression (less cold) to induce *FT1* expression.

Taken together, our results show that conserved flowering genes and temperate grass-specific genes are highly interconnected in a network that integrates plant age, vernalization and photoperiod to mediate a timely flowering response (Fig 8). Our results also show that modulation of the *AP2L* dosage or activity may be a valuable tool to modulate the flowering response of winter cultivars as winters become milder due to climate change.

## Materials and methods

### Plant materials and growth conditions

The tetraploid wheat variety Kronos used in this study has a spring growth habit determined by the *Vrn-A1c* allele (intron 1 deletion), and a reduced photoperiod response conferred by the *Ppd-A1a* allele (promoter deletion). Kronos also has the *AP2L-5A Q* allele, which confers the subcompact spike phenotype and the free-threshing character, and a non-functional *AP2L-5B* gene [66]. TILLING populations of Kronos and Cadenza (hexaploid) mutagenized with ethyl methanesulfonate (EMS) [101] were used to screen for mutants in *AP2L1*. The two selected truncation mutations in *AP2L1* homeologs, and the mutation in the miR172 target site in *AP2L-A1* were confirmed in $M_4$ grains using genome specific primers described in S3 Table.

The mutant $M_4$ plants were crossed at least two times with the parental wild type Kronos to reduce the mutation background before phenotypic analysis. The mutation CAD161 was identified in the hexaploid wheat Cadenza. Since Kronos x Cadenza crosses result in hybrid necrosis, we used a bridge cross between CAD161 mutant to an $F_2$ plant from the cross between the hexaploid line Insignia and the tetraploid Kronos. We then backcrossed the $F_1$ to Kronos twice.

The *vrn-a1* and *vrn1*-null lines [41], and *ap2l5* mutant line K3946 [66] were described before. For all experiments, grains were first cold imbibed for 2–4 days at 4˚C before transferring them to room temperature. The plants were grown in one-gallon pots in PGR15 growth chambers (Conviron) adjusted to 16 h of light (22˚C) and 8 h of darkness (18˚C) (Long Day condition or LD) or 8 h of light (22˚C) and 16 h of darkness (18˚C) (Short Day condition or SD). Intensity of the sodium halide lights measured at plant height was (~260–300 $\mu M\ m^{-2}\ s^{-1}$).

Plants at the second leaf stage were vernalized in one-gallon pots in a cold chamber (Conviron) with temperatures averaging 5˚C and daylength set to 16 h of light / 8 h of darkness. Intensity of the light during the light phase at plant height was 230 $\mu M\ m^{-2}\ s^{-1}$. Vernalization treatments of different length were used in different experiments: In Figs 5G–5K and S9F–S9I, plants were vernalized for 8 weeks; in Fig 6A and 6B, plants were vernalized for 2, 3 and 6 weeks; in Fig 6C–6H, plants were vernalized for 3 and 6 weeks; in S10 Fig, plants were vernalized for 2 weeks; and in S11 Fig, plants were vernalized for 4 weeks.

### qRT-PCR

In all experiments we collected the last fully expanded leaf in the main tiller, and harvested samples 4 hours after lights were turned on, except in Fig 6 where leaves were harvested in the middle of the photoperiod (ZT8). RNA samples were extracted using the Spectrum Plant Total

RNA Kit (Sigma-Aldrich). We followed Protocol A that allows purification of total RNA including small RNA molecules. Total RNA was treated with RQ1 RNase-free DNase (Promega). cDNA synthesis was carried out using SuperScript II Reverse Transcriptase (Invitrogen). mRNAs were reverse transcribed starting from 1 μg of total RNA and using OligodTv primer. Mature miR172 and miR156 levels were determined by stem-loop qRT-PCR as described previously [102], specific stem-loop oligos to mature miR156 and miR172 are described in S3 Table. The reverse primer for the small nucleolar RNA 101 (snoR101), which is the reference used to normalize miRNA in qRT-PCR, was also included in the reverse transcription. RNA extraction and cDNA synthesis for the vernalization time course experiments in Figs 6 and 7 were done as described in [44]. The product from the 1$^{st}$ strand synthesis was diluted 1 in 20, and 5 μl of diluted cDNAs was used in qRT-PCR reaction. Quantitative PCR was performed using SYBR Green and a 7500 Fast Real-Time PCR system (Applied Biosystems). The *ACTIN* gene was used as an endogenous control for mRNAs, and SnoR101 for miRNAs. Primers for the different genes tested are listed in S3 Table.

## Vectors

The coding region of the *AP2L-B1* gene was amplified from Kronos cDNA using primers in S3 Table and cloned into the pENTR vector (Invitrogen). It was then subcloned into the pLC41 vector downstream of the maize *UBIQUITIN* promoter (*UBI$_{pro}$*) with a C-terminal HA tag (henceforth *UBI$_{pro}$:AP2L-B1*). To generate miR156 overexpressing lines, the miR156b,c locus (B genome) was amplified by PCR from Kronos genomic DNA using primers in S3 Table. The PCR product was cloned into pDONR (Invitrogen) and then sub-cloned into pLC41 downstream of *UBI$_{pro}$*. Since *UBI$_{pro}$:miR156b,c* plants were sterile, a LhG4/pOp two component system was developed in wheat to maintain a stable miR156 overexpressing line. LhG4 is a synthetic transcription factor that binds to the synthetic pOp promoter and activates transcription [89]. The *LhG4* coding sequence was cloned into pLC41 vector downstream of the maize *UBI$_{pro}$* to generate the driver vector (*UBI$_{pro}$:LhG4*). The *UBI$_{pro}$* in *UBI$_{pro}$:miR156b*,c vector was replaced by pOp to generate the responder vector (*pOp:miR156*). Finally, *UBI$_{pro}$:LhG4* and *pOp:miR156* plants were crossed, and the phenotypic and molecular analysis were performed in the F$_1$ plants.

To generate the artificial MIM156, the wheat ortholog of the natural miR399-target-mimic *IPS1* gene described in Arabidopsis [103] was synthesized and the sequence complementary to miR399 was replaced by a sequence complementary to miR156. The complete sequence of MIM156 is provided in S11 Fig. The MIM156 construct was cloned into the binary vector pLC41 downstream of the maize *UBI$_{pro}$*. Lines MIM172 and *UBI$_{pro}$:miR172* were described before [66].

## Wheat transformation

Transgenic wheat plants were generated at the UC Davis Plant Transformation Facility (http://ucdptf.ucdavis.edu/) using the Japan Tobacco (JT) technology licensed to UC Davis. Immature embryos from Kronos were transformed using *Agrobacterium EHA105*. Selection of transgenic plants was conducted using hygromycin, and transgene insertion was validated by DNA extraction and PCR.

## Statistical analyses

The raw data and descriptive statistics for all graphs and tables presented in this study are provided in the S1 Data file. Mean comparison were done using Tukey tests that compare all means with each other. In experiments including a wild-type (Wt) control Dunnett tests were

also performed and are included in the S1 Data to provide a more precise estimate of the significance of the differences with the Wt. In all statistical comparisons, homogeneity of variance using Levene's test and normality of residuals using the Shapiro-Wilk test were first tested. If the data did not fit the assumptions of the analysis of variance (ANOVA), power transformations were used to restore them. When power transformations could not satisfy both assumptions, non-parametric Kruskal-Wallis tests were used. All statistical analyses were performed using SAS version 9.4.

Distribution of the data presented with box-plots including individual data points were generated with Excel. The middle line of the box represents the median and the x represents the mean. The bottom line of the box represents the first quartile and the top line the third quartile. The whiskers extend from the ends of the box to the minimum value and maximum value.

## Supporting information

**S1 Table. Gene IDs in CS Ref. Seq. v1.1 for *AP2L* genes in wheat.**
(XLSX)

**S2 Table. miR156 loci in wheat.**
(XLSX)

**S3 Table. Primers used in this study.**
(XLSX)

**S1 Fig. Flowering of Kronos wild type, *UBI$_{pro}$:miR172* and MIM172 lines under SD conditions.** (**A**) Nine-week-old *UBI$_{pro}$:miR172*, wild type Kronos (Wt) and MIM172 plants grown under SD. Scale bar = 10 cm. (**B-C**) Box plots showing days to heading (**B**; n ≥ 7) and the number of leaves produced by the main tiller (**C**; n ≥ 9) in *UBI$_{pro}$:miR172*, wild type Kronos (Wt) and MIM172 plants grown under SD. (**D-G**) Box plots showing expression levels of *VRN1* (**D**), *FT1* (**E**), *VRN2* (**F**), and *PPD1* (**G**) determined by qRT-PCR in the 3$^{rd}$ (L3) and 6$^{th}$ (L6) leaves of *UBI$_{pro}$:miR172*, wild type Kronos (Wt) and MIM172 plants growing under SD. *ACTIN* was used as an internal reference. Data correspond to four independent biological replicates. Different letters above the box plots indicate significant differences based on Tukey tests ($P < 0.05$). Raw data and statistical tests in Data H in S1 Data.
(TIF)

**S2 Fig. Spike phenotypes of *ap2l1* and *ap2l5* mutants.** (**A**) Main spikes and (**B**) central spikelets from wild type Kronos (Wt), *ap2l1*, *ap2l5* and *ap2l1 ap2l5* plants. Red asterisks in *ap2l5* and *ap2l1 ap2l5* spikelets indicate first empty lemmas.
(TIF)

**S3 Fig. Expression levels of *AP2L1* and *AP2L5* in *UBI$_{pro}$:miR172*, wild type and MIM172 plants.** (**A-B**) Box plots showing *AP2L1* (**A**) and *AP2L5* (**B**) expression levels in the 5$^{th}$ leaf of *UBI$_{pro}$:miR172*, wild type Kronos (Wt) and MIM172 plants grown under LD (same biological samples used to quantify miR172 expression in Fig 1B). Transcript levels were determined by qRT-PCR using *ACTIN* as internal reference. Data correspond to four independent biological replicates. Different letters above the box plots indicate significant differences based on Tukey tests ($P < 0.05$). Raw data and statistical tests in Data I in S1 Data.
(TIF)

**S4 Fig. Expression levels of *FUL2* and *FUL3* in *UBI$_{pro}$:miR172*, wild type and MIM172 plants.** (**A-B**) Expression levels of *FUL2* (**A**) and *FUL3* (**B**) determined by qRT-PCR in the 1$^{st}$

(L1), 3<sup>rd</sup> (L3), 5<sup>th</sup> (L5) and 7<sup>th</sup> (L7) leaves of *UBI_pro:miR172*, wild type Kronos (Wt) and MIM172 plants grown under LD. *ACTIN* was used as internal reference. Data correspond to four independent biological replicates. Different letters above the data point for each leaf indicate significant differences based on Tukey tests ($P < 0.05$). The color of the letter corresponds to the color of the genotype. Raw data and statistical tests in Data J in S1 Data.
(TIF)

**S5 Fig. Expression levels of *VRN1*, *VRN2* and *FT1* in the 5<sup>th</sup> leaf of plants segregating for MIM172 and *UBI_pro*:*AP2L-B1*.** (**A-C**) Box plots showing the expression levels of *VRN1* (**A**), *FT1* (**B**) and *VRN2* (**C**) determined by qRT-PCR in the 5<sup>th</sup> leaf of $F_2$ plants segregating for *UBI_pro*:*AP2L-B1* and MIM172 transgenes grown under LD. *ACTIN* was used as internal reference. Data correspond to four independent biological replicates. Different letters above the box plots indicate significant differences based on Tukey tests ($P < 0.05$). Raw data and statistical tests in Data K in S1 Data.
(TIF)

**S6 Fig. Overexpression of miR156b,c locus in Kronos.** (**A**) Scheme showing the cassette including the maize *UBIQUITIN* promoter (*UBI_pro*) and the sequences corresponding to the miR156b,c (B genome) locus. The predicted secondary structure for the miR156b,c sequence is shown below. Note the three stem-loop structures, including the three miR156/miR156* duplexes, corresponding to three miR156 precursors in tandem. (**B**) Wild type Kronos plant (Wt, left) and four independent transgenic T0 lines expressing *UBI_pro*:*miR156b,c* grown under LD. Scale bar = 10 cm.
(TIF)

**S7 Fig. Flowering phenotype of wild type, MIM156, MIM172 and MIM156 MIM172 plants.** (**A-B**) Box plots showing days to heading (**A**; n $\geq$ 9) and the number of leaves produced by the main tiller before the transition to the spike (**B**; n $\geq$ 10) for wild type Kronos (Wt), MIM156, MIM172 and MIM156 MIM172 plants growing under LD conditions. Different letters above the box plots indicate significant differences ($P \leq 0.05$) in pair-wise Kruskal-Wallis tests. Raw data and statistical tests in Data L in S1 Data.
(TIF)

**S8 Fig. Flowering response of winter Kronos to vernalization treatments of different lengths.** (**A-B**) Box plots showing the days to heading (**A**) and the number of leaves produced by the main tiller before the transition to the spike (**B**) for winter Kronos (*vrn-A1*) plants grown under LD conditions without vernalization (NV), with 2 (2WV), 4 (4WV), 6 (6WV) and 8 (8WV) weeks of vernalization. Different letters over the box plots indicate significant differences in Kruskal-Wallis non-parametric pairwise tests at $P < 0.05$. Raw data and statistical tests in Data M in S1 Data.
(TIF)

**S9 Fig. Effect of vernalization and *VRN1* expression on the transcript levels of flowering genes in spring and winter cultivars.** (**A**) Six-week-old spring and winter Kronos plants growing under LD conditions without vernalization. Scale bar = 10 cm. (**B**) Box plots showing the days to heading for spring and winter Kronos plants grown under LD conditions without vernalization (*t*-test, n $\geq$ 6). (**C-E**) Time course of *AP2L5* (**C**), *FUL2* (**D**) and *FUL3* (**E**) expression levels determined by qRT-PCR in spring and winter Kronos plants grown under LD conditions in the absence of vernalization treatment. W1 = week1, 1<sup>st</sup> leaf; W5 = week 5, 7<sup>th</sup> leaf; W10 = week 10, leaves 10<sup>th</sup>-11<sup>th</sup>; W12 = week 12, leaves 13<sup>th</sup>-14<sup>th</sup>; W14 = week 14, leaves 15<sup>th</sup>-16<sup>th</sup>; W16 = week 16, leaves 18<sup>th</sup>-19<sup>th</sup>; W18, week 18, Flag leaves (20<sup>th</sup>-23<sup>rd</sup>). *ACTIN* was used

as internal reference. Data correspond to four independent biological replicates. $* = P \leq 0.05$ in *t*-test comparing expression at the same leaf in spring vs winter plants, except for panel (**E**) where a non-parametric Kruskal-Wallis test was used. (**F-I**) Box plots showing the expression of *AP2L5* (**F**), *VRN-B1* (**G**), *FUL2* (**H**) and *FUL3* (**I**) determined by qRT-PCR in the 7th (L7) leaf of winter Kronos without vernalization, and the 7th leaf of winter Kronos and *vrn1*-null mutants vernalized for 8 weeks and then moved to room temperature for two weeks. Samples from the 7th leaf were collected when the leaves were fully expanded. *ACTIN* was used as internal reference. Data correspond to at least 5 independent biological replicates. (**F**) Same letters indicate lack of significant differences in Tukey tests. (**H**) No expression of *FUL2* was detected for the non-vernalized winter Kronos and the *vrn1*-null samples so no statistical test was performed. (**G and I**) No expression of *VRN-B1* and *FUL3* was detected for the non-vernalized winter Kronos sample, so a *t*-test was performed for the other two samples (*** = $P < 0.001$). Raw data and statistical tests in Data N in S1 Data.
(TIF)

**S10 Fig. Flowering response of MIM172 *UBI*$_{pro}$:*AP2L-B1* winter plants after 4 weeks vernalization.** (**A-B**) Box plots showing days to heading (**A**) and number of leaves produced by the main tiller (**B**) in plants grown under LD conditions with 4 weeks of vernalization. *vrn1*-null = no functional *VRN1* genes, winter Kronos, winter line with MIM172 transgene, winter line with MIM172 and *UBI*$_{pro}$:*AP2L-B1* transgenes. Different letters above the box plots indicate significant differences ($P < 0.05$) in Tukey tests. Raw data and statistical tests in Data O in S1 Data.
(TIF)

**S11 Fig. MIM156 sequence.** The sequence complementary to miR156 is in red.
(TIF)

**S1 Data. Excel File (spreadsheets A to O) including data and statistical analyses supporting figures and supplemental figures.** Data A in S1 Data. Supporting data for Fig 1. Data B in S1 Data. Supporting data for Fig 2. Data C in S1 Data. Supporting data for Fig 3. Data D in S1 Data. Supporting data for Fig 4. Data E in S1 Data. Supporting data for Fig 5. Data F in S1 Data. Supporting data for Fig 6. Data G in S1 Data. Supporting data for Fig 7. Data H in S1 Data. Supporting data for S1 Fig. Data I in S1 Data. Supporting data for S3 Fig. Data J in S1 Data. Supporting data for S4 Fig. Data K in S1 Data. Supporting data for S5 Fig. Data L in S1 Data. Supporting data for S7 Fig. Data M in S1 Data. Supporting data for S8 Fig. Data N in S1 Data. Supporting data for S9 Fig. Data O in S1 Data. Supporting data for S10 Fig.
(XLSX)

## Acknowledgments

We thank Mariana Padilla and Xiaoqin Zhang for excellent technical support. We thank Dr. David Jackson for providing the vectors with *LhG4* and pOp sequences.

## Author Contributions

**Conceptualization:** Juan M. Debernardi, Jorge Dubcovsky.

**Data curation:** Juan M. Debernardi, Daniel P. Woods, Jorge Dubcovsky.

**Formal analysis:** Juan M. Debernardi, Daniel P. Woods, Jorge Dubcovsky.

**Funding acquisition:** Jorge Dubcovsky.

**Investigation:** Juan M. Debernardi, Daniel P. Woods, Kun Li, Chengxia Li.

**Project administration:** Jorge Dubcovsky.

**Resources:** Jorge Dubcovsky.

**Supervision:** Juan M. Debernardi, Jorge Dubcovsky.

**Visualization:** Juan M. Debernardi, Daniel P. Woods, Chengxia Li, Jorge Dubcovsky.

**Writing – original draft:** Juan M. Debernardi, Daniel P. Woods, Chengxia Li.

**Writing – review & editing:** Juan M. Debernardi, Daniel P. Woods, Kun Li, Chengxia Li, Jorge Dubcovsky.

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
