## [Decision Letter · Decision Letter 0]

31 Jan 2022

Dear Dr Dubcovsky,

Thank you very much for submitting your Research Article entitled 'MiR172-APETALA2-like genes integrate vernalization and plant age to control flowering time in wheat' to PLOS Genetics.

The manuscript was fully evaluated at the editorial level and by independent peer reviewers. The reviewers appreciated the attention to an important topic but identified some concerns that we ask you address in a revised manuscript

We therefore ask you to modify the manuscript according to the review recommendations. Your revisions should address the specific points made by each reviewer.

[LINK]

Yours sincerely,

Sarah Hake

Associate Editor

PLOS Genetics

Claudia Köhler

Section Editor: Plant Genetics

PLOS Genetics

Both reviewers are supportive of your work, but raise important considerations. Please deal with all their comments in a revision, especially the concerns of reviewer 2 about carefully describing which leaves you are using and the statistics. I would not suggest leaving out the mir156 data as suggested by reviewer 1.

Reviewer's Responses to Questions

**Comments to the Authors:**

Reviewer #1: This study examines the regulation of the flowering of wheat by two related APETALA2-like transcription factors, AP2L1 and AP2L5 (better known as Q). Two approaches were used to modify activity of AP2-like genes; (1) identification of knockout mutations directly in the genes and (2) altering the activity of a microRNA (miR172) that targets these genes for inactivation, through either transcript cleavage or post-translational mechanisms. The authors show that both of the AP2-like genes delay flowering of wheat. The authors also show that a second microRNA + transcription factor circuit likely acts upstream of the AP2-like genes (MiR156), as is the case in other plants.

This is an interesting study of a worthwhile topic. It will be of interest to readers who are interested in the evolution of the networks that control plant development and those who are interested in crop improvement. The manuscript is clear and easy to follow. The authors present convincing genetic evidence to support the conclusion that AP2-like genes repress flowering of wheat. A few points could be clarified or considered further before publication.

1. Please consider the following point about the impact of AP2L genes on flowering and how this relates to vernalization: when do we conclude that a gene modifies vernalization requirement or vernalization response, versus influencing flowering time per se?

This question is relevant to the section starting at line 530 that suggests that miR172 modulates the vernalization response. Similarly, at line 826, it is suggested that AP2L genes add an additional checkpoint to stop upregulation of FT1 prior to winter and at line 835 it is suggested that AP2L genes could be used to modify the vernalization requirement.

An alternative view is that the AP2-like genes effect flowering time per se. That these genes do not impact the vernalization requirement or response and instead act in parallel by regulating FT1 irrespective of vernalization. And that the magnitude of the effect is dependent on context (genotype and growing conditions). The heading date data presented are consistent with this model. Irrespective of vernalization, over-expression of miR172 (or combined AP2L knockout mutations) accelerates flowering whereas MIM172 delays flowering.

Put differently, would any gene that influences flowering time per se have a similar effect? Would any gene that regulates FT1 activity not elicit a similar pattern?

2. Questions remain with respect to the miR156 section. Plant age might possibly regulate the timing of flowering of non-vernalized winter wheats, which flower only after extended periods. But most of the decrease in miR156 expression seems to happen in the first 5 weeks - are there any changes in winter wheat growth patterns after five weeks? (anything that can be described as a juvenile feature). Also, if the plant age pathway were critical, would you expect the AP2-like genes to be critical for the winter growth habit? Ultimately there was no functional test of whether miR156 can modify the flowering behaviour of non-vernalized winter wheat. Maybe the miR156 section could be omitted.

Reviewer #2: The manuscript describes the analysis of a large number of wheat mutants and transgenic lines to elucidate how the conserved microRNA miR172 and its targets APETALA2-like (AP2L) genes interact with the vernalization pathway involving VERNALIZATION 1 (VRN1), VRN2 and FLOWERING LOCUS T 1 (FT1 =VRN3) genes. The paper includes a lot of interesting data which is clearly described.

I only have a few detailed comments and suggestions.

Methods:

How long was the vernalization treatment?

The authors present a lot of expression data in different figures of the manuscript. In many diagrams, but not in all the authors show expression levels of target genes in different leaves. Please, describe in the Methods section how and when the leaves were harvested, i.e. the 3rd, 5th, 7th leaf were harvested whenever they were fully developed or they were harvested at one time point (when and what time of the day). If the data is available it would be informative to add information on what stage the plants were in when leaves were harvested.

In some figures, there is no information on which leaf/or leaves were used for the expression analysis. For example, Fig. 1H the legend says: Box plots showing the

expression levels of AP2L1 in leaves (H; n = 4).

Fig. 1H, I, J , Fig. 2 etc. Box plots comparing more than two genotypes/conditions: please provide information on sig. differences with a, b, c. etc to distinguish between all different factors (genotypes) rather than only showing comparisons to the WT (asterisks) as it was done for example in Figure 3.

Figure 4: the authors show differences in expression between WT and MIM156 lines for VRN1 and FT1 in leaves 3, 5 and 7. Significant differences were only observed in the 3rd leaf. What was the developmental stage of the shoot apex of the two genotypes at the 3-leaf stage and what were the differences between the genotypes? As the manuscript contains a lot of data, I was wondering if it would be sufficient to show the expression data only for leaf 3 (unless you want to stress here also the expression changes over development for the different genes).

Fig. 5A-F: Here expression levels in winter and spring Kronos in different weeks are displayed. The legend says: W1= week1, 1st leaf; W5= week 5, 7th , leaf; W10= week 10, leaves 10th-11th; etc. I was wondering if in week 5 this was the 5th leaf for the winter Kronos and the 7th leaf for the spring Kronos? In other words, I assume that in W5 the two genotypes were at very different developmental stages. In this context, I was wondering if the sig. tests- comparison of expression levels between genotypes at each time point (W1, W5) makes sense. I think in these figures it would be more important to show sig. difference in expression levels between weeks within each genotype. Also to support the following interpretation

The authors write: In winter lines, we observed a progressive increase in the expression of miR172 with age until week 16 (Fig 5E). Interestingly, we did not observe a parallel downregulation of the transcript levels of AP2L1 (Fig 5F) and AP2L5 (S9C Fig) after W5. This result is similar to the limited effect of UBIpro:miR172 on AP2Ls transcript levels described in S2A and B Fig. After W14, we observed an abrupt downregulation of AP2L1 and AP2L5 that coincided with the up-regulation of VRN1 (Fig 5C), and FUL2 and FUL3 (S9D and E Fig).

Please, add here in the text that it also coincides with the strong upregulation of FT1.

Further, please, provide statistical evidence for the significant down/up regulation of VRN1 and AP2 in week 18. Here again it would be good to provide information on the developmental stage of the spike at W18 (and W16). What do you think could be the role of the AP2 downregulation in W18 for spike development (as far as I can see there are no differences in AP2 expression from W1-W16).

Please, also consider if the expression plots would look much more alike between spring and winter Kronos if you would display W1 and W5 in spring and W1 and W18 in winter as W5 in spring and W18 in winter Kronos may actually be comparable stages.

The figures 5 G-K nicely show that VRN1 indeed regulates AP2 and following figures that AP2 and miR172 modify the vernalization response.

Do you think that AP2 and miR172 modulate VRN1 expression independent of FT1 as suggested in your model? Or would it make sense to include FT1 in your Fig. 8A and B? Some of the links in the model (PhyC, PPD1 etc.) are not derived from data in this paper. You may want to cite the appropriate literature in Fig. 8

**Have all data underlying the figures and results presented in the manuscript been provided?**

Reviewer #1: Yes

Reviewer #2: Yes

PLOS authors have the option to publish the peer review history of their article (what does this mean?). If published, this will include your full peer review and any attached files.

Reviewer #1: No

Reviewer #2: **Yes: **Maria von Korff

---

## [Decision Letter · Decision Letter 1]

20 Mar 2022

Dear Dr Dubcovsky,

We are pleased to inform you that your manuscript entitled "MiR172-APETALA2-like genes integrate vernalization and plant age to control flowering time in wheat" has been editorially accepted for publication in PLOS Genetics. Congratulations!

Yours sincerely,

Sarah Hake

Associate Editor

PLOS Genetics

Claudia Köhler

Section Editor: Plant Genetics

PLOS Genetics

Comments from the reviewers (if applicable):

Thank you for your efforts to address all the reviewers comments.

Reviewer's Responses to Questions

**Comments to the Authors:**

Reviewer #1: The authors have addressed comments and suggestions raised in the initial review.

Reviewer #2: The authors have responded to all my earlier comments.

**Have all data underlying the figures and results presented in the manuscript been provided?**

Reviewer #1: Yes

Reviewer #2: Yes

PLOS authors have the option to publish the peer review history of their article (what does this mean?). If published, this will include your full peer review and any attached files.

Reviewer #1: No

Reviewer #2: No

**Data Deposition**

http://datadryad.org/submit?journalID=pgenetics&manu=PGENETICS-D-22-00019R1

**Press Queries**

---

## [Editor Report · Acceptance letter]

9 Apr 2022

PGENETICS-D-22-00019R1 

MiR172-APETALA2-like genes integrate vernalization and plant age to control flowering time in wheat 

Dear Dr Dubcovsky, 

We are pleased to inform you that your manuscript entitled "MiR172-APETALA2-like genes integrate vernalization and plant age to control flowering time in wheat" has been formally accepted for publication in PLOS Genetics! Your manuscript is now with our production department and you will be notified of the publication date in due course.

With kind regards,

Katalin Szabo

PLOS Genetics

On behalf of:
